# Evaluation of multi-season convection-permitting atmosphere – mixed layer ocean simulations of the Maritime Continent.

Emma Howard[1], Steven Woolnough[2,3], Nicholas Klingaman[2,3], Daniel Shipley[2,3], Claudio Sanchez[4], Simon C. Peatman[5], Cathryn E. Birch[5], and Adrian J. Matthews[6]

[1]Bureau of Meteorology, Brisbane, Australia
[2]National Centre for Atmospheric Science, University of Reading, Reading, UK
[3]Department of Meteorology, University of Reading, UK
[4]Met Office, Exeter, United Kingdom
[5]Institute for Climate and Atmospheric Science, School of Earth and Environment, University of Leeds, Leeds, United Kingdom
[6]Centre for Ocean and Atmospheric Sciences, School of Environmental Sciences and School of Mathematics, University of East Anglia, Norwich, United Kingdom

**Correspondence:** Emma Howard (emma.howard@bom.gov.au)

## Abstract

A multi-season convection-permitting regional climate simulation of the Maritime Continent using the Met Office Unified Model with 2.2-km grid spacing is presented and evaluated. The simulations pioneer the use of atmosphere-ocean coupling with the multi-column $K$ profile parametrisation (KPP) mixed layer ocean model in atmospheric convection-permitting climate simulations. Comparisons are made against a convection parametrised simulation in which it is nested, and which in turn derives boundary conditions from ERA5 reanalysis. This paper describes the configuration, performance of the mean state and variability of the two simulations compared against observational datasets. The models both have minor sea surface temperature (SST) and wet precipitation biases. The diurnal cycle, representation of equatorial waves and relationship between SST and precipitation are all improved in the convection-permitting model compared to the convection parametrised model. The MJO is present in both models with a faster than observed propagation speed. However, it is unclear whether fidelity of the MJO simulation is inherent to the model or whether it predominantly arises from the forcing at the boundaries.

## 1 Introduction

The Maritime Continent (MC) is a key region in the global weather and climate system and a hotspot for tropical convection. Its complex island geography and position among the warmest oceans on Earth lead to a multi-scale concoction of atmospheric convective and dynamical weather systems that aggregate up to a continental scale atmospheric heating pattern (e.g., Yoneyama and Zhang, 2020). The atmospheric response to this heating affects weather and climate across the Earth through modulation of the Hadley and Walker circulations. The Madden-Julian Oscillation (MJO), the leading mode of subseasonal rainfall vari-

ability in the tropics, exhibits large amplitude precipitation anomalies over the MC region (Wheeler and Hendon, 2004), which generate subseasonal teleconnections felt as far afield as the North Atlantic (Lee et al., 2019).

The multi-scale nature of convection in the MC poses a particular challenge for Earth system models, particularly those with parametrised convection (Neale and Slingo, 2003). This results in systemic errors in climate models such as model biases, misrepresentations of diurnal cycles, and reduced variability across time-scales (e.g., Baranowski et al., 2019). The advent of large-domain, high resolution convection-permitting model simulations with horizontal grid spacing finer than 6km provides a new tool to examine the distribution and variability of MC convection. In a convection-permitting simulation, the

grid spacing is small enough to allow individual deep convective clouds to be explicitly resolved. However, at coarser resolution these convective processes, their triggering, and their associated transports of mass, momentum, heat, and moisture must be parametrised, often with significant shortcomings. For example, misrepresentation of the diurnal cycle (Slingo et al., 2003) has follow on effects to the propagation of the MJO (Ling et al., 2019). Thus, deficiencies in the parametrisation of these processes, including their dependence on multi-scale interactions, may be identified by comparing convection parametrised

and permitting simulations.

     Multi-year convection-permitting simulations of the MC region have been used with great effect to explore the scale interactions of the MC and the wider warm pool region. Birch et al. (2016) analysed a 10 year continuous simulation of the western MC (Sumatra and Peninsular Malaysia) at 4.5 and 12 km grid-spacing and established that better-resolved scale interactions such as rain-forming sea-breeze convergence lead to an improved representation of the precipitation signal of the MJO. Wei

et al. (2020) found that the relationship between the MJO and diurnal precipitation is varied within the region due to the complexities of island geometry and topography using a 3-km model. Vincent and Lane (2018) demonstrated the varied roles of stratiform and convective diabatic heating through phases of the MJO in a 4 km WRF-based simulation of the whole MC region, and King and Vincent (2018) examined the ENSO teleconnection to the MC in these same simulations, first described by (Vincent and Lane, 2017). The importance of mesoscale off-shore propagating waves in driving diurnally propagating pre-

cipitation has been demonstrated using convection-permitting models over Sumatra by Love et al. (2011) and over New Guinea by Hassim et al. (2016). Hagos et al. (2016) studied the importance of the diurnal cycle to the MC barrier effect on MJO propagation by comparing convection-permitting simulations with and without a diurnal cycle of incoming solar radiation. Across these studies, an understanding of processes across a wide range of length-scales have been made possible by the inclusion of kilometre-scale convection.

Advances in computational ability allow for increasingly large simulations with larger grids and longer time periods to be performed. This is advantageous on numerous counts. Firstly, as the kilometer-scale models are still not able to fully resolve storm processes such as convective updrafts, the representation of precipitation remains imperfect and takes on a 'blobby' characteristic (e.g. Stratton et al., 2018). Between grid-spacings of 4 km and a few 100 m, any increase in resolution tends to improve the representation of convection (Potvin and Flora, 2015; Stein et al., 2015). For example, (Argueso et al.,

2020) demonstrated that simulations with explicit convection with a 2-km grid-spacing outperforms coarser resolutions of both explicit and parametrised convection at simulating the timing of the diurnal precipitation peak over the Maritime Continent, and attributed this to improved resolution of cloud structure. Secondly, the use of increasingly large domains is possible. While

in smaller domains the influence of boundary forcing inhibits feedbacks from convective scale behaviour back onto the large-scale flow, in a larger domain, scale interactions between convection and increasingly large-scale atmospheric features are made possible. Finally, increased computational resource increases the total number of days that may be simulated, allowing for the simulation of a more complete range of temporal variability, including intraseasonal, seasonal and interannual, to be sampled and for longer time-scale feedbacks to develop.

The convection-permitting simulations described above all use atmosphere-only models with observed sea surface temperatures (SSTs) prescribed as the lower boundary conditions over the ocean. This means that air-sea interactions, for example between the ocean mixed layer and convection, are not resolved. However, as the warm ocean forcing drives the strong MC convection, it follows that the ability of the SSTs and mixed-layer depth to respond to atmospheric forcing is likely to be important for the nature of regional convection. As summarised by DeMott et al. (2015), some studies have suggested that ocean-atmosphere coupling may sustain and encourage propagation of the MJO. Although many general circulation models have a poor representation of the MJO, it has been shown that coupled models can exhibit better MJO propagation than their uncoupled counterparts (e.g., Hirons et al., 2015; Shelly et al., 2014).

A summary of coupled regional climate simulations of the MC region has been provided by Xue et al. (2020). All multi-year simulations they considered have been run at grid-spacings of 15km and coarser, such that atmospheric convective processes must be parametrised. They found that ocean-atmosphere coupling was able to improve biases in SST (compared to standalone ocean models) and sometimes improve precipitation biases, but that poor representation of unresolved processes such as clouds, convection and the complex dynamics of the Indonesian throughflow caused substantive model deficiencies. In shorter simulations, Thompson et al. (2019) found that ocean coupling improved forecast quality on a case-study basis when simulating cold surges in the South China Sea, while (Thompson et al., 2021) found a good agreement between a 4.5km atmosphere-ocean coupled forecast system and local observations in near surface ocean fields.

In order to properly resolve air-sea interactions, a coupled model with an oceanic component is required. However, this raises issues of its own: ocean models often feature warm SST biases in the Maritime Continent region of order 1-2° C (e.g., Wang et al., 2022), which exacerbate rainfall biases and draw the simulated climatology further away from reality. Ocean models also require a lengthy spin-up period, adding to the technical complexity of the modelling frameworks. A compromise presented by Hirons et al. (2015) is the *K* profile parametrisation multi-column mixed layer ocean model (hereon referred to as KPP), which uses the Large et al. (1994) vertical mixing parametrisation to simulate mixing of temperature and salinity, and parametrises the effects of ocean dynamics and other processes through a combination of relaxation and flux correction. This configuration has been shown to limit SST biases to less than 0.5° (Hirons et al., 2015). The KPP mixed layer ocean has the further advantage that it is not limited by the Courant-Friedrichs–Lewy (CFL) condition, as it does not simulate oceanic advection. This means that it can be run with both a finer vertical grid-spacing than many other ocean models, which improved its representation of the mixed layer, and also with a long timestep, making it computationally efficient.

This paper presents 30 months of convection-permitting and parametrised simulations over the full MC region using 2-km and 12-km grid-spacing, respectively. The simulations have been performed over ten December - February seasons, when tropical modes of variability such as ENSO and the MJO are at their strongest. The Met Office Unified Model (MetUM)

atmospheric model has been used, coupled for the first time in a convection-permitting climate simulation to a KPP mixed layer ocean. The MetUM is a seamless framework for weather forecasting and climate prediction (Brown et al., 2012). Limited area domains are used, with boundary conditions derived from ERA5 reanalysis. This dataset adds to a collection of convective-scale research simulations generated using the MetUM, including CASCADE (Pearson et al., 2014) and CP4-Africa (Stratton et al., 2018).

These simulations are expected to capture interactions between different length-scales and model components including: land and sea breezes, feedbacks between precipitation and SST; deepening of the ocean mixed layer following rainfall; topographic triggering and offshore propagation of convection; convective organisation; convergence lines; and diurnal cycles of precipitation and coastal sea surface temperatures. Larger-scale circulation systems including the MJO, equatorial waves, tropical cyclones, cold surges in the South China Sea and dry intrusions from the Australian continent occur during the course of the simulations, and feedbacks between these processes and MC convection are expected to be captured by the simulations.

## 2  Data and Methods

### 2.1  Experimental Design

Two configurations of the atmospheric model coupled with ocean mixed layer model have been used, to compare the effect of parametrised convection against explicit convection. In the first configuration, denoted MC12, the model is run with parametrised convection on a limited area domain of a standard MetUM grid configuration known as N1280, which has a zonal spacing of 0.140625° and a meridional spacing of 0.09375°. This corresponds to approximately 12 km at the equator. The MC12 domain spans from 85°E to 160°E and 20°S to 20°N, encompassing Malaysia, Indonesia, Papua New Guinea and the southern islands of the Phillippines. In the second configuration, denoted MC2, the model is run with explicit convection and has a grid spacing in both the zonal and meridional direction of 0.02 degrees. This corresponds to approximately 2 km at the equator. The MC2 model is nested within the MC12 domain, with a 5 degree buffer zone, such that the MC2 domain spans from 90°E to 155°E and 15°S to 15°N. Maps presented in the paper show the MC12 domain with the MC2 domain marked by a black box.

Ten seasons have been selected to span a range of conditions, which form a climatology containing the major modes of inter-annual climate variability that impact Southeast Asia. Seasons have also been chosen to coincide with major observational campaigns in the region, associated with the Years of the Maritime Continent. Each season runs for three months from the 1st of December through to the 28th of February the following year, with November run but not analysed for the purpose of model spinup. Table 1 indicates the years selected and the phases of considered modes of variability - El Niño Southern Oscillation (ENSO), the Indian Ocean Dipole (IOD), the Quasi-Biannual Oscillation (QBO), and the amplitude of the MJO - for each of the selected seasons.

| Season | ENSO | IOD | QBO (50 hPa) | MJO Activity | Field Campaign |
|--------|------|-----|--------------|--------------|----------------|
| 2003-04 | Neutral | Positive | Easterly | High | |
| 2005-06 | La Niña* | Negative | Easterly | Low | |
| 2007-08 | La Niña | Negative | Easterly | High | |
| 2009-10 | El Niño | Positive | Westerly | Low | |
| 2012-13 | Neutral | Positive | Easterly | High | |
| 2014-15 | Neutral | Negative | Easterly (T) | Low | |
| 2015-16 | El Niño | Positive | Westerly | High | Mirai |
| 2016-17 | Neutral | Negative | Westerly | Low | |
| 2017-18 | La Niña | Positive | Westerly | High | Mirai |
| 2018-19 | El Niño* | Positive | Easterly (T) | High | ELO |

**Table 1.** Phase of climate variability indices during selected seasons for simulations. The second column is based on Niño 3.4 SST index and indicates when this index exceeds ±0.5°C. This threshold is weaker than the official ENSO threshold, and * indicates years where the full ENSO criteria have not been met. The second threshold indicates when the Dipole Mode Index is positive or negative. The Quasi-Biannual Oscillation (QBO) is shown in column 3, which displays whether the winds above Singapore at 50 hPa are Easterly or Westerly. Transition seasons where the DJF season in question is not near a local extremum are indicated by the symbol (T). The MJO activity column indicates whether or not at least two-thirds of the days in the season feature an active MJO with RMM amplitude exceeding 1. The 'Field Campaign' column indicates any observational campaign that overlaps with the simulation. Specifically, these are the radiosonde campaigns conducted off the coast of Bengkulu with the Mirai research vessel (Yokoi et al., 2017) and the Equatorial Line Observations (ELO) Seaglider campaign (Azaneu et al., 2022).

## 2.2 Atmospheric Model and Land Surface

The atmospheric model component is the MetUM. Model science settings for MC12 are as per the Global Atmosphere 7
(GA7) configuration (Walters et al., 2019) and includes a mass flux parametrisation for atmospheric convection based on Gregory and Rowntree (1990). Lateral boundary conditions are obtained from ERA5 (Hersbach et al., 2020) and update every six hours. The inner nested domain, MC2, uses the tropical version of the Regional Atmosphere and Land 2 (RAL2T) science configuration (Bush et al., 2023). A comparison of the differences between atmosphere and land settings in the two nests is provided in Table 2. In the MC2 configuration there is no deep convection parametrisation, as the resolution is high enough
for deep convective processes to be explicitly simulated. Therefore, a comparison between the two sub-domains allows for an examination of the effect of explicit vs parametrised convection.

The land surface configurations of each nest follow the GA7 and RAL2T specifications accordingly. Both use the Joint UK Land Environment Simulator (Jules) with 4 soil levels (Best et al., 2011). Soil moisture is initialised using a land-surface reanalysis created by forcing the JULES land surface model with data from the Japanese Meteorological Agency's atmospheric
analysis (JRA-55). This dataset was generated to initialise the UK Met Office Global Seasonal forecast system version 5 (GloSEA5, MacLachlan et al., 2015) hindcast system from April 2019 onwards. Soil moistures are interpolated from 1 degree

resolution to the model grids, and so a month of model spin-up is allowed at the beginning of each simulation to allow the soil moisture to equilibriate to this resolution change. In order to simplify the coupling between the atmosphere and the KPP-ocean, coastal tiling is not used, so each grid-cell is classified either as fully land or fully ocean.

In order to test the role of air-sea coupling, an additional atmosphere-only simulation has been run for a single season (2015-16) only. This simulation used the same model configuration as the core coupled model experiments, but used 3-daily updating foundational OSTIA SST as a sea-surface boundary condition.

|  | MC2 | MC12 |
|---|---|---|
| Atmos Zonal Grid Spacing | $0.022°$ ($\sim$ 2km) | $0.14°$ ($\sim$ 15km) |
| Atmos Meridional Grid Spacing | $0.022°$ ($\sim$ 2km) | $0.09°$,($\sim$ 10km) |
| Atmos Horizontal grid-cells (x,y) | (3200,1500) | (538,428) |
| Vertical levels | 80 levels with 40 km top | 70 levels with 80 km top |
| Latitude range | $15°$ S - $15°$ N | $20°$ S- $20°$ N |
| Longitude range | $90°$ E-$155°$ E | $85°$ E-$160°$ E |
| Science Settings | RA2T (Bush et al., 2019) | GA7 (Walters et al., 2019) |
| Initial condition | ERA5 | ERA5 |
| Boundary condition | MC12 | ERA5 |
| Timestep | 1 min | 3 min |
| Convection Scheme | No | Yes |
| Large-scale Cloud Scheme | PC2 (Wilson et al., 2008) | PC2 (Wilson et al., 2008) |
| Graupel and lightning Diagnosis | Yes | No |
| Land Surface Scheme | Jules (Best et al., 2011) | Jules (Best et al., 2011) |
| Land Surface Initialisation | As per GloSEA5 | As per GloSEA5 |
| Land Surface Spinup | 1 month | 1 month |

**Table 2.** Summary of configuration setup of atmosphere and land models for MC2 and MC12.

## 2.3   KPP Ocean

The oceanic component is comprised of the multi-column *K* profile parametrisation (KPP) mixed layer ocean model (Large

et al., 1994) with coupling configured following Hirons et al. (2015). The KPP is comprised of a set of independent 1-D KPP column models, each separately coupled to one or more atmospheric grid-points by the Ocean Atmosphere Sea Ice Soil (OASIS) coupler (Valcke, 2013). The KPP model simulates vertical mixing of salinity and temperature, but not large-scale advection. Because of this, CFL limitations do not apply, and so a relatively long timestep and fine vertical grid-spacing may be applied.

In the present configuration, the KPP model is run on a stretched grid with a maximum depth of 250 m, with 70 levels and with vertical spacing in the top layer of 0.8 m. In both atmospheric configurations described above, the KPP ocean is run on

a subset of the standard N1280 grid, aligned with the MC12 atmospheric mass-grid. For MC12, this results in a one-to-one coupling between ocean grid-points in the atmosphere and in the KPP model. For MC2, fluxes into each ocean point are derived from the average of the atmosphere points directly above it, while atmospheric grid cells receive fluxes from the nearest oceanic neighbour on the N1280 grid. The KPP model is both updated and coupled to the atmosphere every hour. The land mask for both models is set to match that of the atmospheric component of MC12, derived from the International Geosphere-Biosphere Program dataset (Townshend, 1992), while the bathymetry is derived from that of the NEMO reanalysis as described below, with a maximum depth of 300 metres applied. The KPP solves a discretised version of equations (1) and (2) below:

$$\frac{\partial T}{\partial t} = \underbrace{\frac{\partial}{\partial z}\left(-\overline{w'T'}\right)}_{\text{KPP mixing}} + \underbrace{\delta_{z=0}Q}_{\text{surface heat flux}} + \underbrace{\delta_{z=0}\frac{\partial}{\partial z}Q_{sw}J(z)}_{\text{penetrative solar}} + \underbrace{\frac{1}{\tau}\left(T_{\text{ref}} - T\right)}_{\text{Relaxation}} + \underbrace{F_T}_{\text{Forcing}} \tag{1}$$

$$\frac{\partial S}{\partial t} = \underbrace{\frac{\partial}{\partial z}\left(-\overline{w'S'}\right)}_{\text{KPP mixing}} - \underbrace{\delta_{z=0}SF}_{\text{freshwater flux}} + \underbrace{\frac{1}{\tau}\left(S_{\text{ref}} - S\right)}_{\text{Relaxation}} + \underbrace{F_S}_{\text{Forcing}} \tag{2}$$

Here, $T$ and $S$ are temperature and salinity, $Q$ is the surface heat flux excluding the contribution of net shortwave radiation, $Q_{sw}$ is the net downwards shortwave radiation, J(z) is a modified Jerlov double exponential (Jerlov, 1976), SF is the surface salinity flux, $\tau$ is a relaxation timescale and $T_{ref}$ and $S_{ref}$ are reference temperatures and salinities. $F_T$ and $F_S$ represent the externally imposed forcing described below. $\overline{w'T'}$ and $\overline{w'S'}$ represent the subgrid-scale vertical fluxes of temperature and salt, which are parametrised using the KPP mixing scheme. The Jerlov double exponential uses the parameters of Jerlov water type IB, but is modified in shallow water such that it reaches 0 at the bathymetry and is differentiable with continuous first derivative. These equations provide two mechanisms to prevent model drift and compensate for the lack of ocean dynamics: relaxation and forcing. These are used in tandem to ensure that the KPP ocean maintains a realistic mean state without damping out crucial modes of intra-annual variability, as described by Hirons et al. (2015) and below. The relaxation mechanism restore temperatures and salinities to the reference state, damping oceanic variability in the process over a time-scale $\tau$. The forcing mechanism simply prescribes a predetermined source or sink of heat ($F_T$) or salt ($F_S$). On sub-seasonal timescales, forcing is preferable to relaxation as it preserves intraseasonal variability in the ocean-atmosphere coupling. Both mechanisms rely on the existence of the reference temperatures and salinities.

### 2.3.1 Reference temperatures and salinities

For consistency with the ERA5 atmospheric boundary conditions, sea surface temperatures were obtained using HadISST2.0 up to 2007 and daily foundational OSTIA SST (Good et al., 2020) from 2007 onwards. Salinity profiles and the variation of the temperature profiles with depth were both obtained from the NEMO reanalysis, a global ocean eddy-resolving reanalysis using the NEMO Ocean Model with $1/12°$ horizontal grid spacing and 50 vertical levels produced by CMEMS (Lellouche et al., 2021). All input data were first smoothed in time with a 7-day running mean and interpolated with an area-weighted regridder to the KPP model grid. To generate the full temperature relaxation profiles, the skin temperature was first removed from the NEMO reanalysis product by setting all temperatures in the top 5 metres to the NEMO reanalysis foundational SST,

defined as the temperature at a depth of 5 metres. Then, at each horizontal grid-cell the NEMO vertical profiles were shifted by the difference between the OSTIA and the NEMO reanalysis foundational SSTs. This ensured that the surface temperatures would match OSTIA and that the vertical derivatives would match the NEMO reanalysis.

### 2.3.2 Correction technique

A heat and salt correction technique is applied using a reference daily temperature and salinity derived from ocean reanalysis products to prevent model drift. These corrections balance the surface flux biases from the coupled atmosphere and compensate for the lack of oceanic advection in the KPP model. In order to generate appropriate forcing, the MC12 model is run twice in two different modes. This multi-step process is illustrated schematically in Fig. 1. This figure shows the input datasets (rectangles) used in each simulation stage (circles) and computation step (diamonds).

- Firstly (green) the reference temperatures and salinities are calculated as described in Sect. 2.3.1.

- Secondly (purple) the MC12 configuration is run with no forcing ($F_S = F_T = 0$) and with a fast relaxation time-scale of 15 days. This time-scale was demonstrated by Hirons et al. (2015) to achieve a balance between countering SST drifts and allowing surface fluxes to adjust to the presence of a coupled ocean. These simulations were run for a period of 15 seasons initialised each year between 2003 and 2017. Each simulation began on the 1st of November and ended on the 15th of March the following year. These relaxation simulations maintain low sea surface temperature biases, but their internal variability is damped and the ocean prognostic variables are unable to evolve independently due to the strong relaxation.

- Thirdly, (blue) the 15-year climatological daily mean of the relaxation increments applied during these simulations was computed and smoothed with a 30-day rolling mean, labelled $\overline{C_T}$ and $\overline{C_S}$ in Fig. 1. These resultant climatologies contain edge effects associated with the rolling means up until the 15th of November each year - this was deemed acceptable since all of November is disregarded as the land-surface spin up. These fields contain the corrections necessary to maintain low SST biases achieved in the relaxation simulations.

- Finally (red), the seasonal climatological means of the heat and salt corrections were applied in the production run simulations as $F_T = \overline{C_T}$ and $F_S = \overline{C_S}$. Application of the climatological corrections as a forcing term allows the production runs to have low SST biases without the model's internal variability being damped.

Interannual oceanic variability was provided to these production runs using an additional slow relaxation to $T_{\text{ref}}$ and $S_{\text{ref}}$ with a 90 day time-scale. These production runs were run for the 10 DJF seasons described in Section 2.1 and comprise a convection parametrised climatology of the Maritime Continent. The 15-year correction climatology period was chosen to surround the 10-year study period such that if a new season is to be rerun in the future, generation of a new correction climatology will not be necessary.

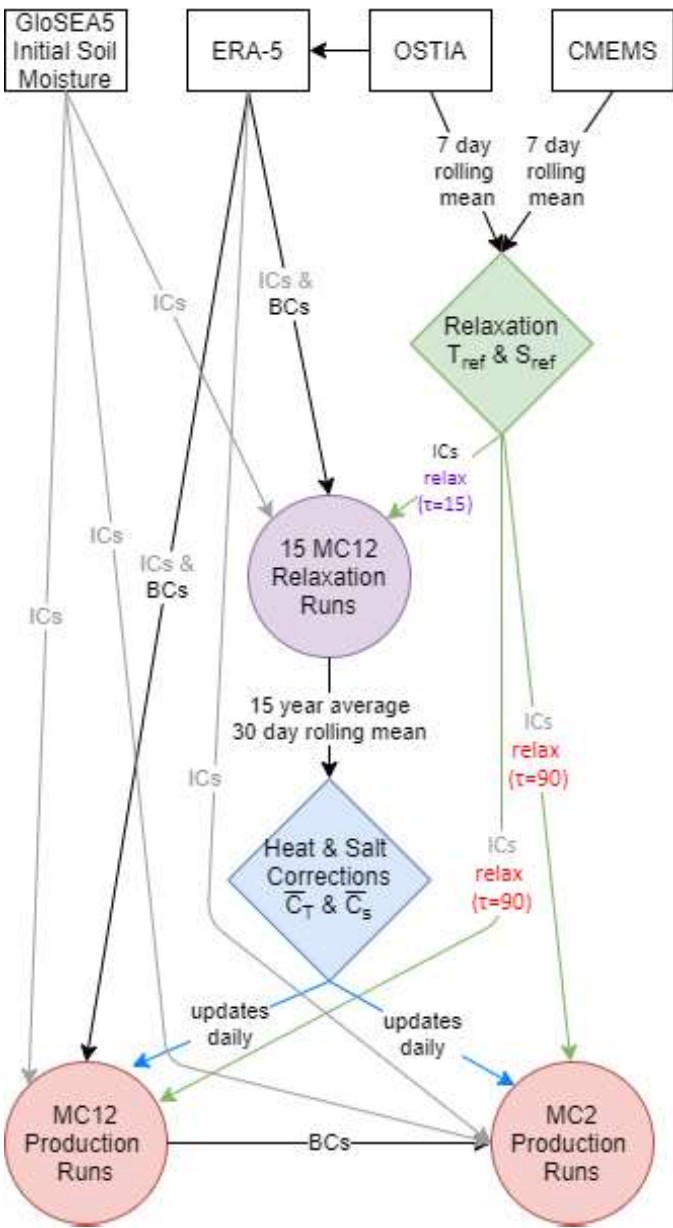

**Figure 1.** Schematic showing the flow of information between datasets and simulations in the generation of reference datasets, heat & salt corrections, initial conditions (ICs) and boundary conditions (BCs). Rectangles: datasets, diamonds: major calculation steps, circles: simulations.

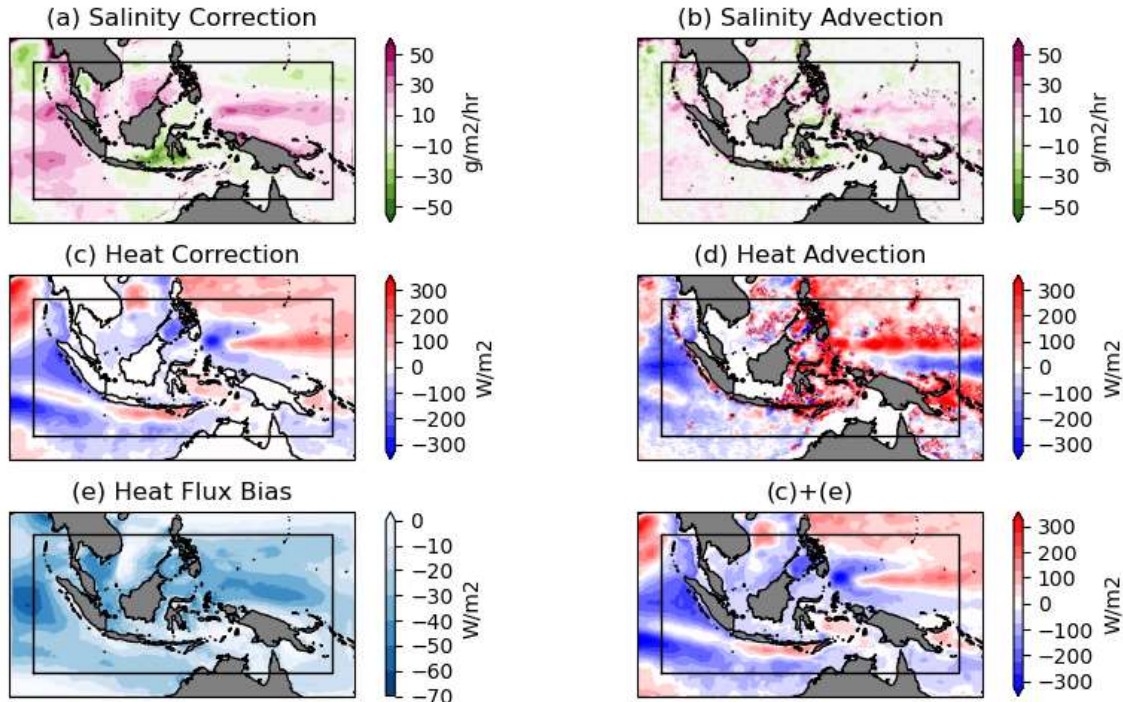

**Figure 2.** Comparison between salt and heat corrections (panels a and c), mean-flow salinity and heat advection in a reference reanalysis (panels b and d) and the surface heat flux bias (panel e). The mean state advection terms are calculated as $-\nabla \cdot \overline{\mathbf{u}}\overline{X}$. The lower right panel represents the sums of panels c and e, which is the component of the heat correction that may be expected to balance panel d.

### 2.3.3 Validation Check

To confirm that the heat and salt corrections are reasonable, the surface and vertically integrated seasonal mean values of $\overline{C_T}$ and $\overline{C_S}$ are compared in Fig. 2 with the mean advection computed directly from the monthly mean NEMO reanalysis temperature, salinity and currents and with the relaxation run surface biases, using ERA5 as a reference. Vertical advection was calculated by integrating the continuity equation with the assumption that the vertical velocity is zero at the surface. From this it can be seen that the KPP salinity correction closely matches the large-scale features of the NEMO reanalysis salinity advection, although the correction term is much smoother. Comparisons between panels (d) and (f) demonstrate a qualitative correspondence between the NEMO reanalysis heat advection and the component of the KPP model heat correction that does not balance the surface heat flux bias. This component of the KPP heat correction has the same sign as the NEMO reanalysis heat advection term in many locations, particularly with the cooling effect of equatorial upwelling on the west coast of Sumatra and with advective warming along the North Equatorial Current in the Western Pacific. The KPP heat correction is roughly half the magnitude of the NEMO reanalysis mean-state heat advection. This may be due to the 15-day relaxation timescale, or the absence of the eddy component of the heat advection in the NEMO reanalysis calculation.

| Variable | Reference dataset | Model | Mean | Bias | RMSE | Std. dev. | Bias | RMSE |
|---|---|---|---|---|---|---|---|---|
| Precipitation [mm/day] | GPM-IMERG | MC2 | 8.43 | 1.1 | 3.2 | 3.01 | 0.466 | 1.47 |
| | | MC12 | 8.5 | 1.2 | 2.97 | 2.42 | -0.112 | 1.34 |
| | MSWEP | MC2 | 8.43 | 1.35 | 3.41 | 3.01 | 1.25 | 1.77 |
| | | MC12 | 8.5 | 1.45 | 3.01 | 2.42 | 0.67 | 1.33 |
| SST [°C] | OSTIA | MC2 | 28.8 | -0.0188 | 0.144 | 0.337 | 0.0296 | 0.0867 |
| | | MC12 | 28.8 | 0.0231 | 0.205 | 0.349 | 0.0421 | 0.0973 |
| TOA outgoing longwave [W/m$^2$] | NOAA daily OLR | MC2 | 225 | 0.19 | 8.06 | 11.4 | 1.14 | 2.46 |
| | | MC12 | 225 | -0.2 | 14 | 13.1 | 3.07 | 4.03 |
| TOA outgoing shortwave [W/m$^2$] | NCEP-NCAR reanalysis | MC2 | 121 | -20.8 | 28.1 | 11.5 | 3.43 | 5.16 |
| | | MC12 | 120 | -21.9 | 29.4 | 11.7 | 3.59 | 5.34 |
| U850 [m/s] | ERA5 reanalysis | MC2 | -1.5 | -0.69 | 1.1 | 1.47 | 0.226 | 0.798 |
| | | MC12 | -1.71 | -0.835 | 1.13 | 1.28 | 0.0232 | 0.235 |
| V850 [m/s] | | MC2 | -1.14 | -0.0659 | 0.514 | 0.511 | -0.0383 | 0.328 |
| | | MC12 | -1.14 | -0.0671 | 0.42 | 0.652 | 0.0996 | 0.199 |
| Zonal mean uplift [Pa/s] | | MC2 | -0.0309 | -0.00397 | 0.0192 | 0.00942 | 0.00122 | 0.00452 |
| | | MC12 | -0.0319 | -0.00493 | 0.00841 | 0.00858 | 0.000378 | 0.00183 |
| Zonal mean specific humidity [g/kg] | | MC2 | 5.59 | 0.0571 | 0.188 | 0.184 | -0.00709 | 0.047 |
| | | MC12 | 5.77 | 0.237 | 0.331 | 0.171 | -0.02 | 0.0477 |
| Zonal mean air temperature [K] | | MC2 | 263 | 0.21 | 0.441 | 0.298 | 0.00569 | 0.0329 |
| | | MC12 | 264 | 0.24 | 0.573 | 0.293 | 0.000282 | 0.038 |
| Zonal mean U [m/s] | | MC2 | -3.09 | 0.00449 | 0.923 | 1.12 | 0.00532 | 0.119 |
| | | MC12 | -3.07 | 0.0225 | 1.2 | 1.08 | -0.0398 | 0.188 |
| Zonal mean V [m/s] | | MC2 | 0.124 | 0.0151 | 0.392 | 0.321 | 0.0152 | 0.0725 |
| | | MC12 | 0.138 | 0.029 | 0.576 | 0.315 | 0.00896 | 0.126 |

**Table 3.** Summary of domain-averaged time-mean values and interannual standard deviations of variables discussed in the text, as well as their respective biases and RMS errors relative to reference datasets. All values are given to 3 significant figures. All averages are computed over the MC2 domain only. Averages of zonal-mean fields are calculated for pressures less than 100 hPa.

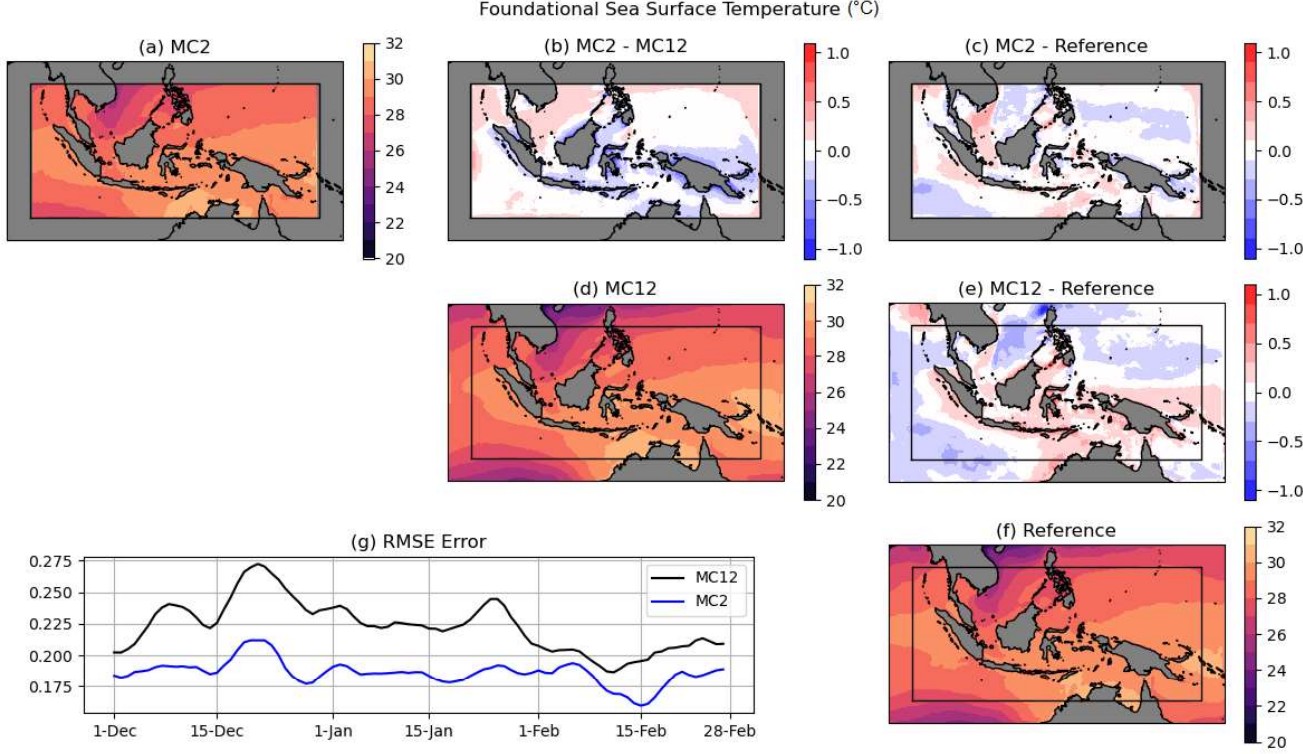

**Figure 3.** Foundational sea surface temperatures and biases (°C). Subplots along the diagonal indicate MC2, MC12 and reference SST respectively. The fourth model level from the surface, with a depth of 4.5 metres, was chosen to represent the foundational sea surface temperature. Upper off diagonal subplots show difference plots. Bottom left: Root mean square error across MC2 domain of each model compared to the 10-year mean reference SST dataset on each date.

The computationally expensive MC2 domain was only run once, in production mode with heat and salt corrections $\overline{C_T}$ and $\overline{C_S}$ derived from the MC12 relaxation runs. This framework assumes that the surface flux biases generated using the different atmospheric frameworks (MC2 and MC12) are sufficiently similar that heat and salt corrections generated using MC12 are appropriate for MC2. This assumption is validated in Supplementary Figure S1, which demonstrates that the surface heat flux bias is very close across the relaxation, MC2 and MC12 runs, with differences of less than 10 $\mathrm{Wm}^{-2}$ compared to an overall bias of 20-40 $\mathrm{Wm}^{-2}$ (relative to ERA5).

## 2.4 Evaluation Datasets

This paper evaluates the climatological mean-state and modes of variability simulated in the MC2 and MC12 simulations. To do so, several observational and reanalysis-based references are drawn on as benchmarks. ERA5 (Hersbach et al., 2020) reanalysis is used to evaluate the atmospheric state, including temperature, winds, specific humidity and uplift on pressure levels. Only the high resolution, deterministic ERA5 realisation is used. GPM-IMERGv06B (Huffman et al., 2018) is used

to evaluate precipitation on diurnal, daily and monthly timescales. GPM-IMERG has been demonstrated to provide accurate estimates of precipitation up to the 95th percentile (Da Silva et al., 2021).We used only the "precipitationCal" product, which is satellite observations including correction to rain gauge. Finally, the ocean temperature reference dataset constructed by combining the OSTIA SST (Good et al., 2020) and the NEMO reanalysis (Lellouche et al., 2021), described in section 2.3.1, is used to evaluate climatological ocean temperature.

## 3    Mean State

This section considers the mean state of the atmosphere and mixed-layer ocean across the 10 simulation years;Table 3 summarises the domain-mean values, as well as biases and RMS errors relative to reference datasets, of variables considered in this section. Figure 3 shows the foundational mean sea surface temperature in the two models and the reference SST dataset averaged across the simulations. Here, the fourth model level from the surface, with a depth of 4.5 metres, was chosen to be the most comparable to the foundational reference SST, free of diurnal variability. MC12 shows a warm bias of around 0.2K in off-shore coastal regions and a cold bias in the Western Pacific (panel e). MC2 shows the greatest difference from MC12 in coastal regions (panel b), where its performance is much improved over MC12(panel c). MC2 biases are generally less than 0.3K everywhere and are maximised in the South China Sea. The RMS error (panel h) indicates that the MC12 SST drifts tend to grow in December but are stable through January and February, while the MC2 SST drifts are constant throughout.

That the SST biases are smaller in MC2 was unexpected, as the relaxation runs to determine the KPP flux corrections were only performed for MC12. A full exploration of the mechanisms behind this difference is beyond the scope of this paper, but a preliminary analysis suggests that the changes in SST biases between MC2 and MC12 are broadly consistent with changes in the surface heat flux and these changes are also consistent with changes in the mixed-layer depth (not shown). This is especially clear around the islands of the Maritime Continent, with cooler SST and deeper mixed layers associated with lower heat fluxes into the ocean. To properly understand the differences in the SST biases would however require a detailed analysis of the time evolution of the biases and would be regionally sensitive.

Both models exhibit a wet bias compared to the GPM-IMERG climatology averaged over the same period, as indicated by Figure 4. This is consistent with other high resolution coupled models in the Maritime Continent region (e.g., Roberts et al., 2019), with both parametrised and explicit convection, especially in DJF (Liu et al., 2023). The wet bias over the ocean is stronger in MC12 than MC2 and has a typical magnitude of around 3 mm/day (panels c, e). Both models also have strong wet biases over high orography, and MC12 has a dry bias over lowlands in Sumatra, Borneo and Java.It is common for convection-permitting models to exacerbate wet or dry biases compared to the same model run with parametrised convection; this is certainly true for the Met Office Unified Model (e.g. Muetzelfeldt et al., 2021), and in general there is no systematic improvement in mean precipitation bias for convection-permitting models over -parametrised models (Prein et al., 2015). Strong boundary effects can be seen near the equator in MC2, where localised wet biases are present 3° from the edges of the domain, consistent with other regional convection-permitting modelling over the MC (Jones et al., 2023).

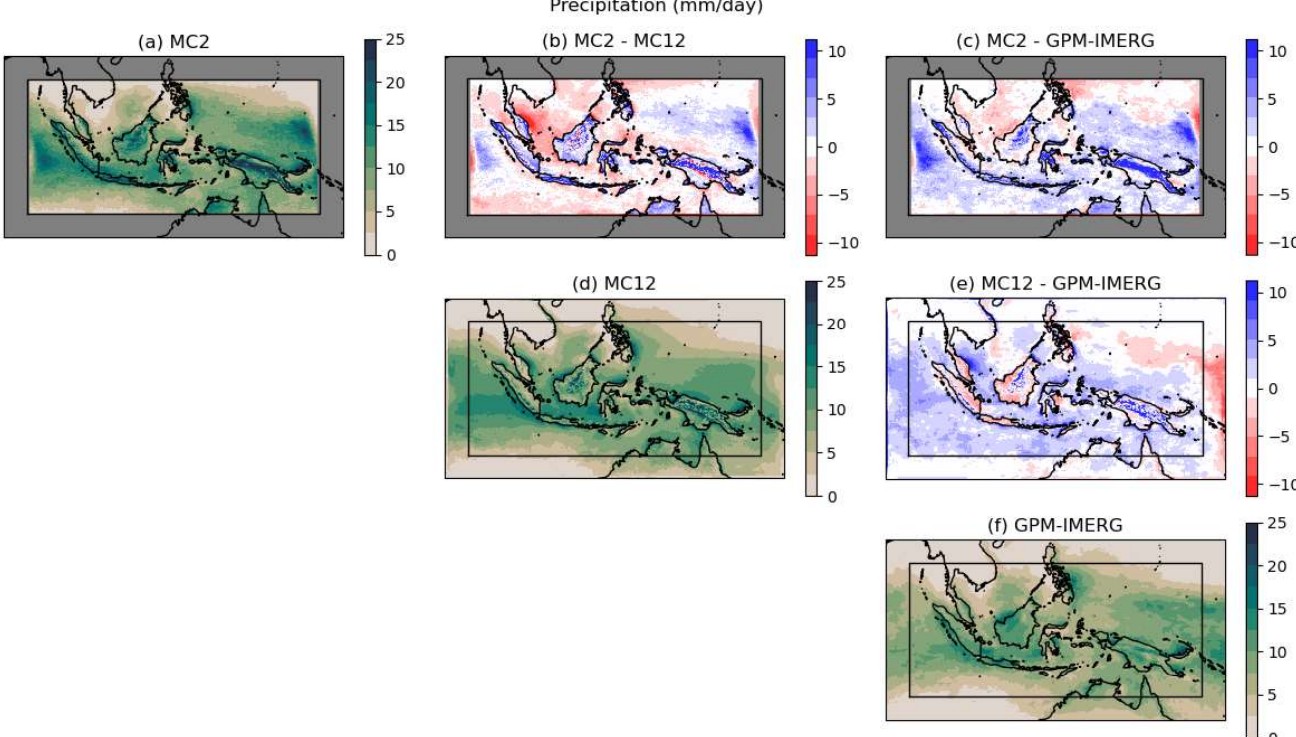

**Figure 4.** Precipitation totals and biases (mm/day). Subplots are arranged as per Figure 3. The reference dataset is GPM-IMERG.

The timing of diurnal precipitation is indicated in Figure 5. In panel a, MC2 shows the complex diurnal cycle that is characteristic of the Maritime Continent, with rainfall over land peaking in the mid-afternoon and propagating offshore overnight. Comparison with GPM (panel c) indicates that MC2 is able to represent the timing of the maximum rainfall very skilfully. Typically of parametrised convection (e.g. Slingo et al., 2003), MC12 (panel b) rains too early over land and coastal regions, particularly in low-lying regions such as the south east of Sumatra, where the peak occurs at 12pm.

Due to the sparsity and incompleteness of rain gauge observations over the MC (e.g. Figs. 2, 3, B4, B8 of Lewis et al., 2019), there is potentially large observational uncertainty in precipitation over our domain of interest. This is especially true over areas with high orography, where the MC2 and MC12 biases are worst compared to GPM-IMERG, and of course over the ocean, where there is a general wet bias in both models. It is therefore beyond the scope of this paper to perform a direct comparison with gauge data. However, to give some insight into how observational uncertainty may impact the model evaluation, we have compared the precipitation climatology of both simulations to the "Past" product from the Multi-Source Weighted-Ensemble Precipitation (MSWEP) v2.8 dataset, which combines satellite and rain gauge observations with reanalysis (Beck et al., 2019). Mean biases relative to MSWEP were similar both in magnitude and spatial pattern to those relative to GPM-IMERG (Supplementary Figure S2). Differences in diurnal cycle phase have been found to be negligible between different precipitation

products over the Maritime Continent, especially at the 3 hour temporal sampling rate of many observational datasets (see e.g. Supplementary Figure 7 of Dong et al., 2023).

As proxies for cloud cover, we also compared the outgoing longwave and reflected shortwave from both model simulations against observations (see Supplementary Figures S3 and S4). We find that in both MC2 and MC12 the biases are broadly consistent with typical GCM biases, and that the biases (and RMS errors) in MC2 tend to be reduced compared to MC12, suggesting a generally better representation of clouds in MC2 (consistent with a higher-resolution convection-permitting simulation).

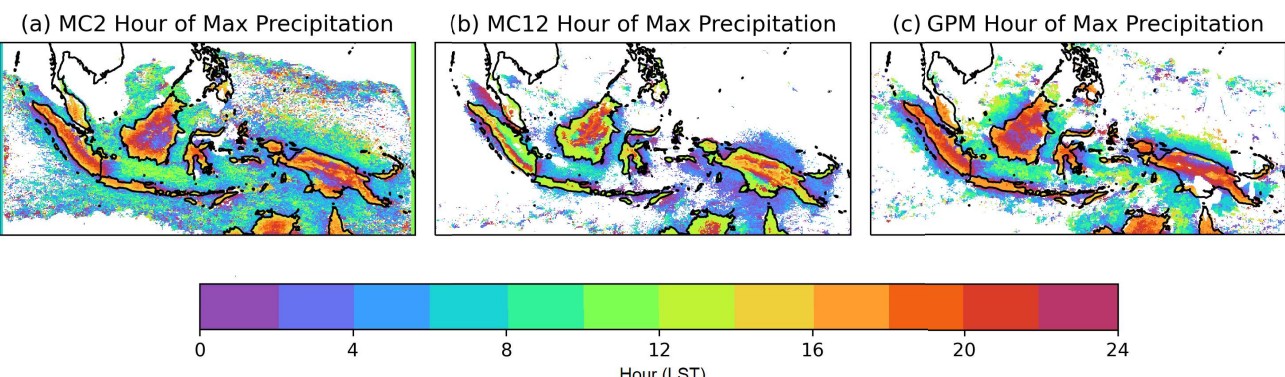

**Figure 5.** Hour of maximum precipitation during mean diurnal cycle in local solar time. White regions indicate where the mean diurnal amplitude is less than the mean rainfall rate, and where the overall DJF climatological rainfall is less than 5mm/day. Left: MC2, Centre: MC12, Right: GPM-IMERG.

Figure 6 shows that the mean state vertical structure of the atmosphere is generally similar to that of the driving reanalysis, with humidity and uplift biases of order 10% of their overall values. The core uplift zone is offset to the south of the equator due to the timing of the simulations during the Austral Monsoon. MC12 has a positive humidity bias at 600 hPa on both sides of the tropical core (panel e). This bias is absent from MC2 (panel c). Both models show an increase in deep uplift in the northern hemisphere compared to ERA5, centred on 300hPa at about $2°$ N.

Overall, both models do reasonably well at simulating the mean-state climatology of the Maritime Continent. SST biases are low due to the careful configuration of the KPP model, where a negative heat flux bias into the ocean of around 30 W/m2 is managed by the heat corrections. Despite these heat corrections being optimised for MC12, MC2 has lower SST biases than MC12. The large-scale vertical structure of the atmosphere is well constrained by the boundary conditions. An overall wet-bias of order 1.2 mm/day is present in both models.

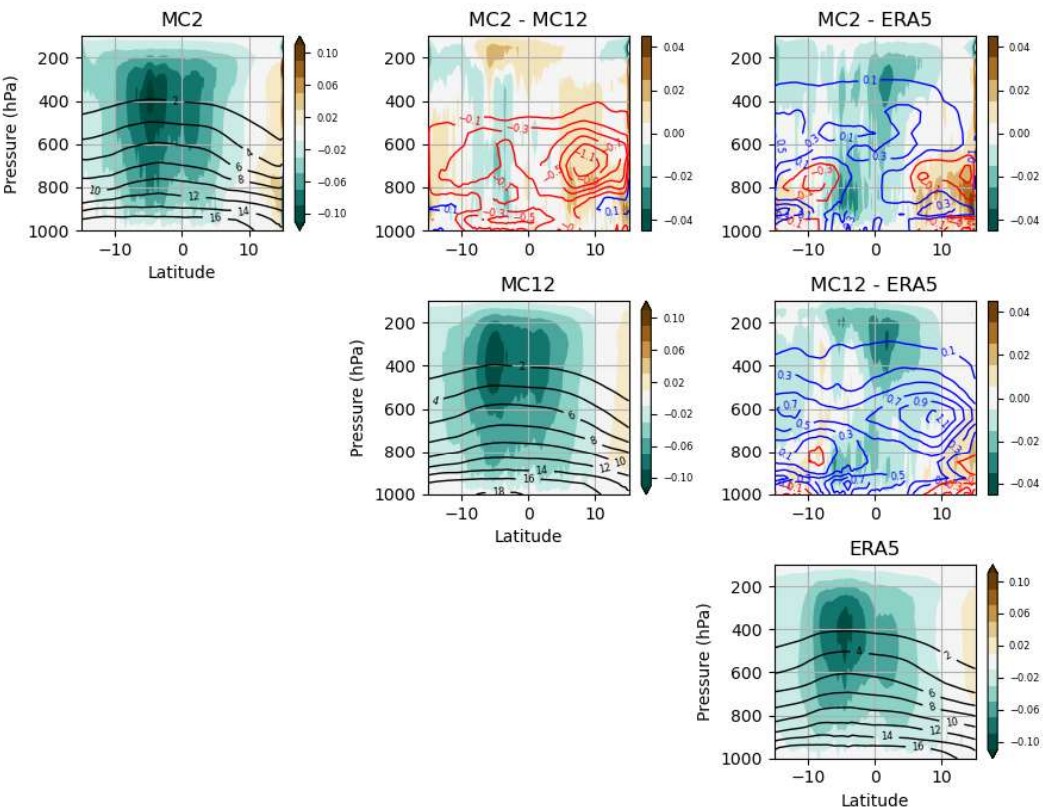

**Figure 6.** Cross sections of vertical atmospheric structure. Zonal means have been taken across the MC2 domain (90° –155° E). Filled contours: uplift and subsidence (Pa/s), lines: specific humidity (g/kg). On difference plots, red lines indicate negative values and blue lines positive values. Subplots are arranged as per Figure 3. The reference dataset is ERA5 reanalysis.

## 4 Modes of Atmospheric and Oceanic Variability

It was demonstrated in Section 3 that MC2 and MC12 possess reasonable mean-state climatologies that compare well with ob-
servational datasets. This section examines the representation of atmospheric and SST variability in both models. Sub-seasonal
variability in the form of equatorial waves and the MJO is considered first, followed by interannual variability, including ENSO.

### 4.1 Subseasonal Variability

To understand the representation of sub-seasonal variability in the MC2 and MC12 models, and the degree to which it is
inherited from the driving boundary conditions, a case-study approach is first considered. Figure 7 shows Hovmöllers of pre-
cipitation across three latitude transects for a selected month, December 2015. At this stage of the integration, the model is
already spun-up and has been free-running away from the boundaries of the domain since the start of November. Common
large-scale precipitating features can be seen across the two models and the observational products. Most prominent are Ty-

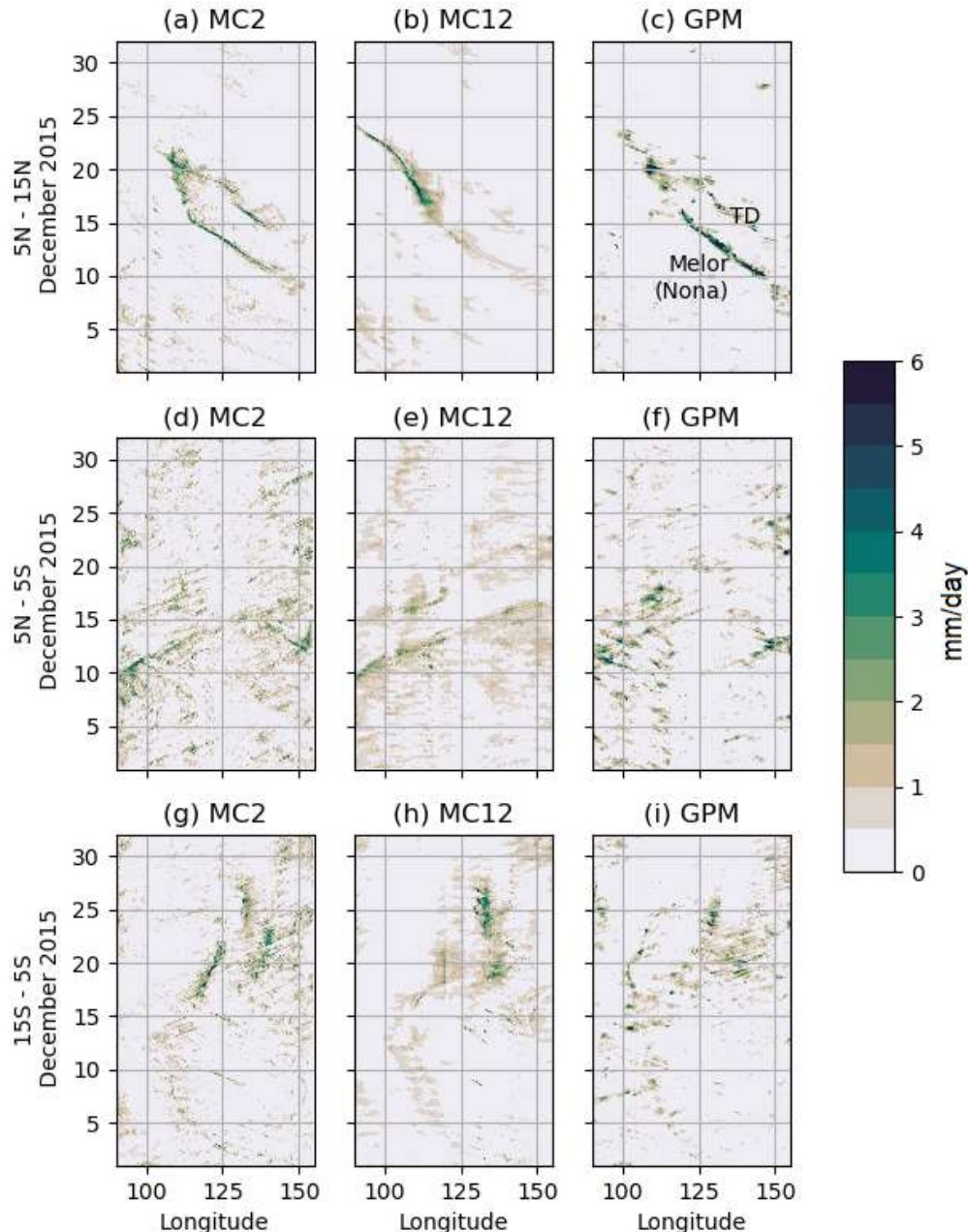

**Figure 7.** Hovmöller plots of precipitation averaged across latitude bands in mm/day. Top row: 5°N - 15°N, middle row: 5°S - 5°N, bottom row: 15°S - 5°S. Left: MC2, centre: MC12, right: GPM-IMERG.

phoon Melor (Nona) and an unnamed tropical depression, which dominate the northern domain (7a-c). The typhoon entered

the northern transect from the east on the 10th of December and propagated through to the Philippines on the 15th, where it stalled for a few days before the precipitation signal merged with that of the tropical depression. The system stalls most noticeably in the MC2 model, while it precipitates the least in the MC12 model, particularly between 120° and 150° E. The tropical depression is also barely apparent in MC12.

In the equatorial transect (panels d-f), several Kelvin waves propagate through the transect. The strongest of these enters at the western boundary on the 10th of December in both models and GPM-IMERG. The precipitation signal persists in both MC2 and MC12 for longer and further than in GPM-IMERG observations. The signal is more diffusive in MC12. In the southern transect (panels g-h), a period of enhanced convection occurs between the 18th and the 27th of December, focused particularly on the eastern half of the transect. This coincides with a burst of high MJO activity in phases 5 and 6. Overall, there is a high degree of similarity between the two models and the driving observations at large length-scales, particularly when phenomena propagate into the domain from the boundaries. This is sufficient to identify the same phenomena across models. However, smaller scale properties of these phenomena differ in a systematic manner, allowing for the impacts of the different model configurations to be examined.

The representation of the MJO and equatorial waves is examined by considering lagged regressions of daily latitude-averaged and band-pass filtered precipitation and 850 hPa zonal winds. Time series at a fixed base-point are correlated with time-series at each longitudinal point of the domain to show signals propagating through the domain from this basepoint. Using a low-pass filter of 20 days is a common technique to demonstrate MJO propagation (e.g., Hirons et al., 2015). We apply this approach in Figure 8 and also include a high-pass filter, separated into eastward and westward propagating components, to additionally observe the propagation of features that propagate at the speeds of equatorial waves. Eastward and westward signals are separated using 2-d Fourier filters in longitude and time. The third and fifth rows of Figure 8 additionally show a modification where base point time-series is sourced from in observations and correlations are calculated against model output. These rows are intended to determine the degree to which the modelled propagating features are in phase with corresponding observed features that enter the domain through the eastern and western boundaries. Thus, rows 3 and 5 test whether the models have MJO-like and equatorial wave-like variability at the same time as observations, while rows 2 and 4 demonstrate whether the models get the correct propagation structure whenever MJO-like and equatorial wave-like variability is present in their own realities, potentially out of phase from observed timing. Propagation speeds in metres per second for the wind signals were calculated by fitting a linear least-squares regression to the local maximum in longitude at each time-step along the central green ridge-lines respectively. These are indicated in the bottom right corner of each subplot.

Propagation of low frequency MJO-like features in the 850 hPa zonal winds from 100E is very similar between observations and models (Figure 8, black contours, column 1). Eastward propagation likely to be associated with the MJO dominates, and the dynamical signal remains coupled with the driving observations. The +0.6 wind contour (i.e. the third solid black contour) extends to 150° E in panel (d) but only to 135° E in panel (j), indicating that the propagating wind signal is stronger in MC2 than MC12. The rainfall signal experiences a jump behaviour bypassing the Maritime continent in panels (a), (g) and (m), indicative of the vanguard effect (Peatman et al., 2014). This is much less clear when the correlation is performed within the model output, although a hint is present in panel (d). Overall, eastward MJO-like propagation of zonal winds and precipitation

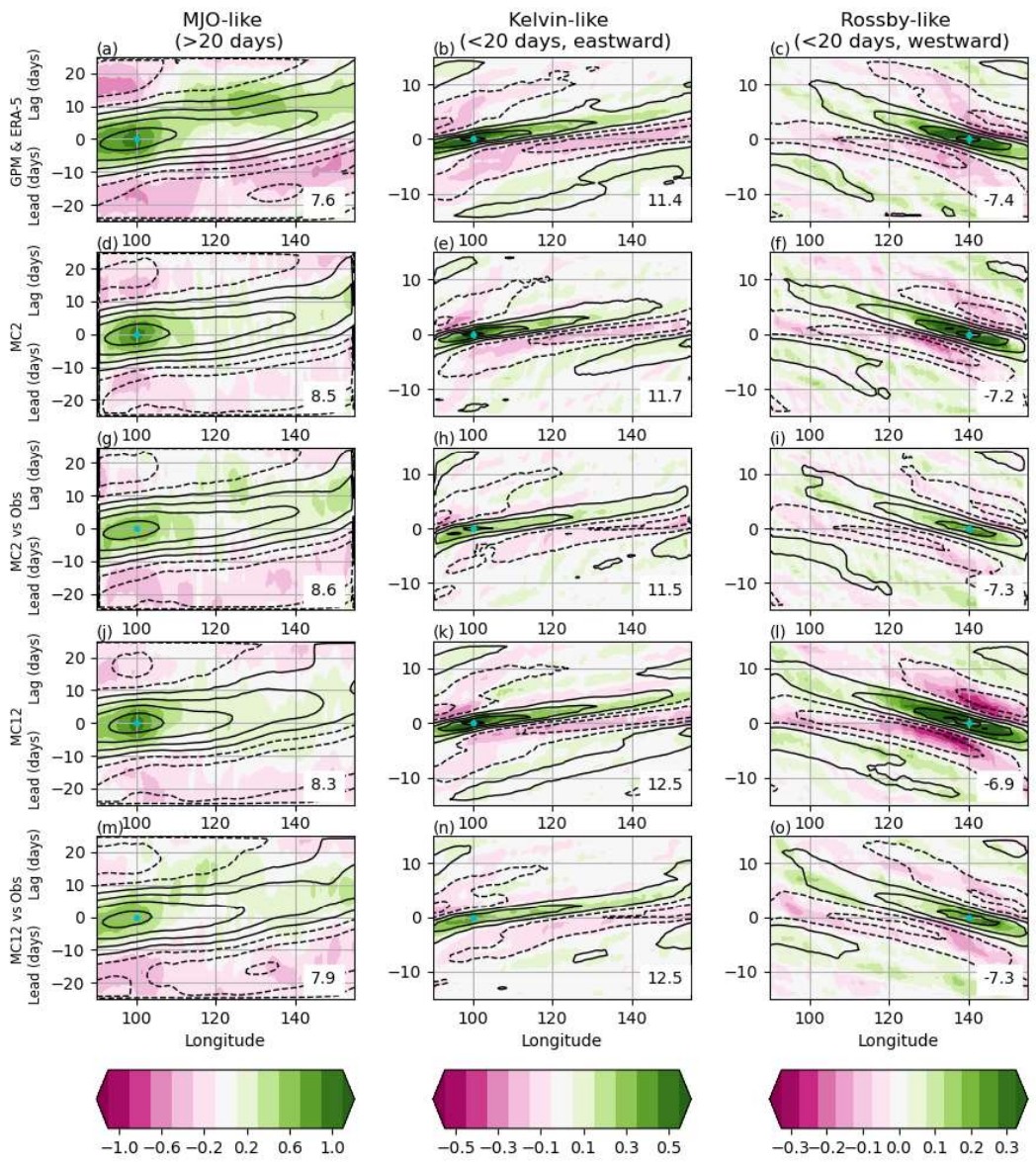

**Figure 8.** Lagged auto-regressions of filtered precipitation (colours) and winds (line contours) with base-point indicated by blue dot. Left: 20 day low-pass filter, centre: 20 day high-pass filter, eastward-propagating signals only, right: 20 day high-pass filter, westward-propagating only. Rows 1, 2 and 4 show observations/reanalysis, MC2 and MC12 respectively. Rows 3 and 5 show correlations of MC2 and MC12 respectively with the base-point derived from the observational products. The number in the bottom right gives an indication of propagation speed based on the slope of the ridge passing through the blue dot, in m/s.

is improved compared to similarly configured global MetUM simulations (e.g., Hirons et al., 2015) simply due to the presence
of the forced dynamical signal.

The eastward signal is similar in magnitude and propagation speed between the simulations and the observational products. The correlation between the simulations and the observational base-point is positive but weak (Figure 8h and 8n), which suggests that although Kelvin waves are driven by the lateral boundaries, they develop independent behaviour and move partially out of phase with the observations. Westward propagating disturbances in MC12 exhibit broader and stronger auto-correlations, suggesting that they are overestimated in MC12. This is consistent with an overestimation of the signal intensity of equatorial Rossby waves and tropical cyclones often observed in MetUM simulations with parametrised convection (Feng et al., 2020). The central propagating signals in panels (c) and (f) are very similar, suggesting that MC2 generates and propagates Rossby waves quite accurately.

Comparisons of rows 2 and 4 of Figure 8 with rows 3 and 5 indicates the degree to which MC2 and MC12 are in phase with their observationally-derived boundary conditions. The magnitude of all four MJO-like signals in the left column(panels d, g, j and m) are similar, suggesting that MJO variability is generally closely coupled to the boundary conditions. The model-based equatorial wave-like signals (panels e, f, k and l) show approximately double the signal amplitude of the corresponding waves correlated with observations (panels h, i, n and g), suggesting that there is some phase alignment between equatorial waves between the models and the driving boundary conditions, but also considerable scope for independent evolution.

Overall, both MC2 and MC12 are able to generate their own equatorial wave-like variability, and these waves are able to evolve independently of the boundary forcing. MC2 produces a better climatology of equatorial waves than MC12, with improved magnitudes and propagation speeds. Although MJO-like variability is present in both models, it is largely controlled through the boundary conditions. Dynamical propagation is too fast and the rainfall signal is overly weak.

The rainfall signal of the MJO is examined further in Figure 9, which shows composite rainfall anomalies in each phase of the MJO in the MC2 and MC12 models. Following the results of Figure 8, which showed that the modelled MJO matches the observed MJO quite closely, this analysis does not take the divergence between the real-world and modelled MJO phase into account, and instead bases composites on the observational RMM MJO index. Both models feature similar rainfall anomalies over the ocean, but diverge over land. MC2 shows a much stronger signal over land, for example over Borneo in phases 2 and 5, and an improved vanguard effect in phases 2, 3, 5 and 6 (Peatman et al., 2014). An illustrative case can be seen near the Malay Peninsula where in phases 3 and 8, strong anomalies are present that are onshore in MC2 and offshore in MC12.

## 4.2   Interannual variability

The observed interannual standard deviations of 850 hPa winds, sea surface temperature and precipitation, together with their modelled biases, are presented in colours in Figure 10. In each case, standard deviations are comparable but are generally slightly overestimated in both models compared to the observations(this is also true for the top-of-atmosphere shortwave and longwave radiation fluxes; see Supplementary Figures S5 and S6). MC2 and MC12 both show excess variability in low-level winds to the south of Sumatra, representing the strength of the Austral monsoon westerly winds along their southern edge. SSTs show greatest variability in the Indian Ocean and in the South China Sea in both models and observations. This variability stretches through to the Java sea in the model, but not in the observed SSTs. Inter-annual precipitation variability is strongest along the edge of the tropical rain-band and is overestimated in MC2, particularly over land, and underestimated

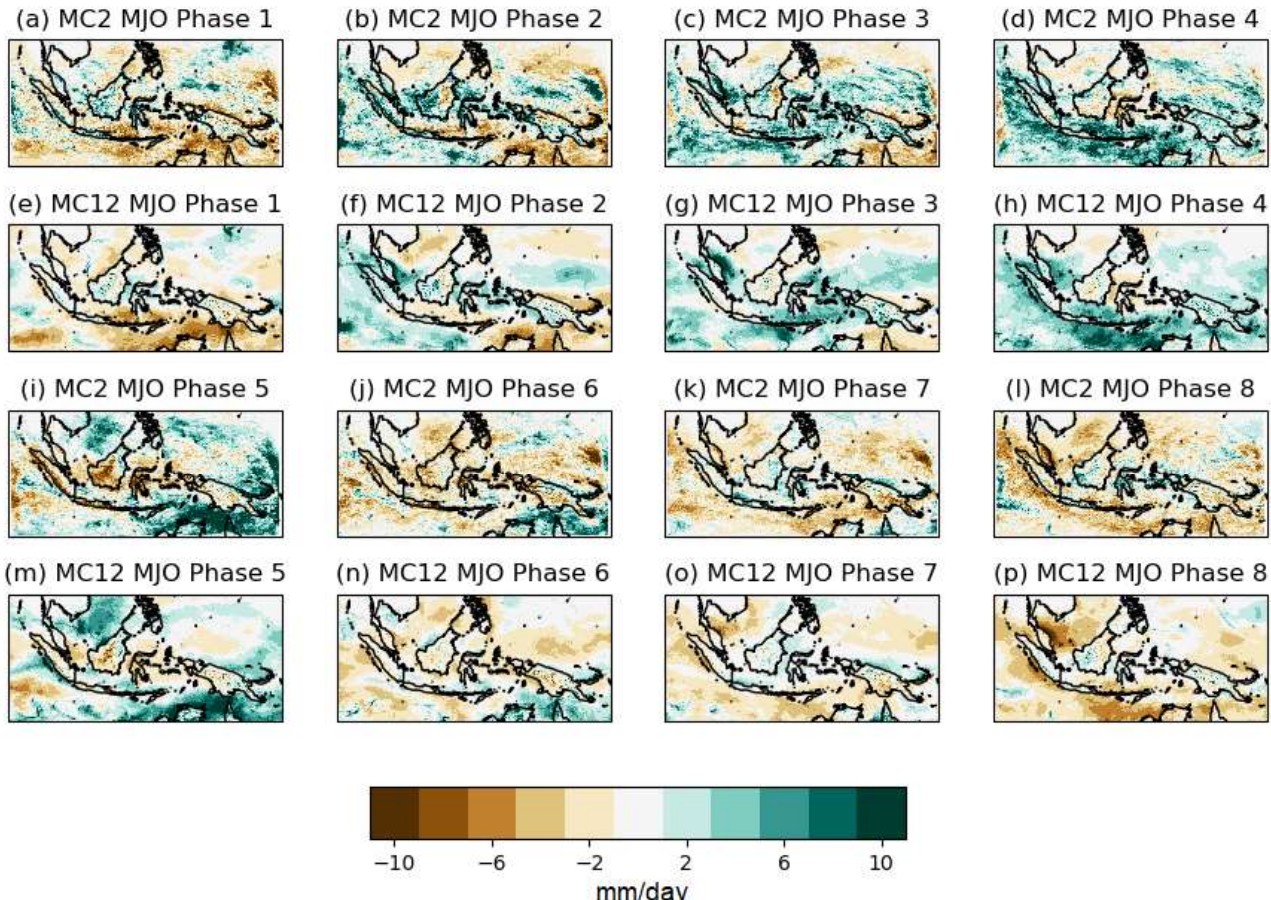

**Figure 9.** Anomalous rainfall composites by MJO phase (mm/day). Rows 1 and 3: MC2, rows 2 and 4: MC12.

over the ocean in MC12. The locations of enhanced precipitation due to boundary effects in MC2 are also evident as spurious variability hotspots.

Given the uncertainty in precipitation observations, we again compared the model interannual standard deviations of precipitation to MSWEP (Supplementary Figure S7). GPM-IMERG has notably more interannual variability over ocean than MSWEP, but the variability over land is comparable, likely due to the rain-gauge correction of both datasets. This strengthens the conclusion that MC2 overestimates the interannual variability of precipitation, but weakens confidence in the sign of the bias over ocean for MC12.

Figure 11 extends the analysis of interannual standard deviations to vertical cross-sections of the zonal mean atmosphere. Variations of specific humidity and uplift are most pronounced at around 5°N, reflecting fluctuations in the northward extent

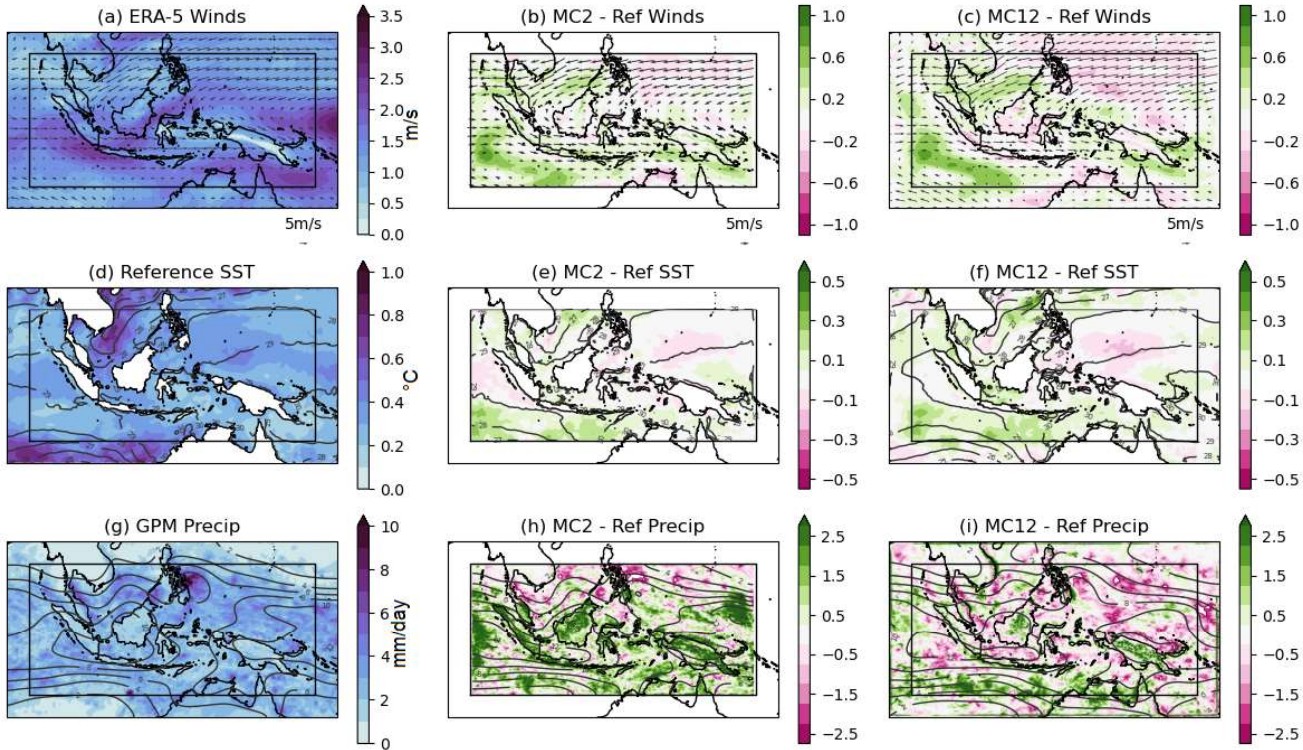

**Figure 10.** Interannual standard deviations of key variables and their differences from the observations and reanalysis. These standard deviations have been computed by first calculating seasonal means for each of the 10 DJF seasons, and then taking the standard deviation. The standard deviation of the two dimensional vector wind field is taken as the square root of the sums of the variances of each component. Top: 850 hPa winds, middle: foundational sea surface temperature, bottom: precipitation. Climatological mean fields composed in arrows (top) or contours (middle and bottom). A Gaussian smoothing filter with a 20 MC12 grid-point radius has been applied to the precipitation climatology (line-contours only) for readability. The standard deviation of winds is taken as the square root of the sums of the variances of the zonal and meridional wind components. Left: observational/reanalysis based reference, centre: MC2 bias from observations, right: MC12 bias from observations.

of the inter-tropical convergence zone (ITCZ). The maximum specific humidity variability, located on the northern side of the
domain, is reduced in both models compared to observations. MC12 shows excessive variability in shallow convection across the low-levels of the domain which is mitigated in MC2, where shallow convection is partially resolved. Both MC2 and MC12 show a reduction in the interannual variability at the core of the convective hotspot near 2°S. Zonal wind standard deviations indicate too much variability in the depth of the westward winds in MC2, and the southern extent in MC12.

    The previous two figures demonstrate that interannual variability is present at levels that match well with reanalysis and
observational products across the seasons being studied. However, they do not give an indication of whether this variability is in phase with remote drivers of inter-annual variability. To address this, the left panels of Figure 12 show latitudinal profiles of

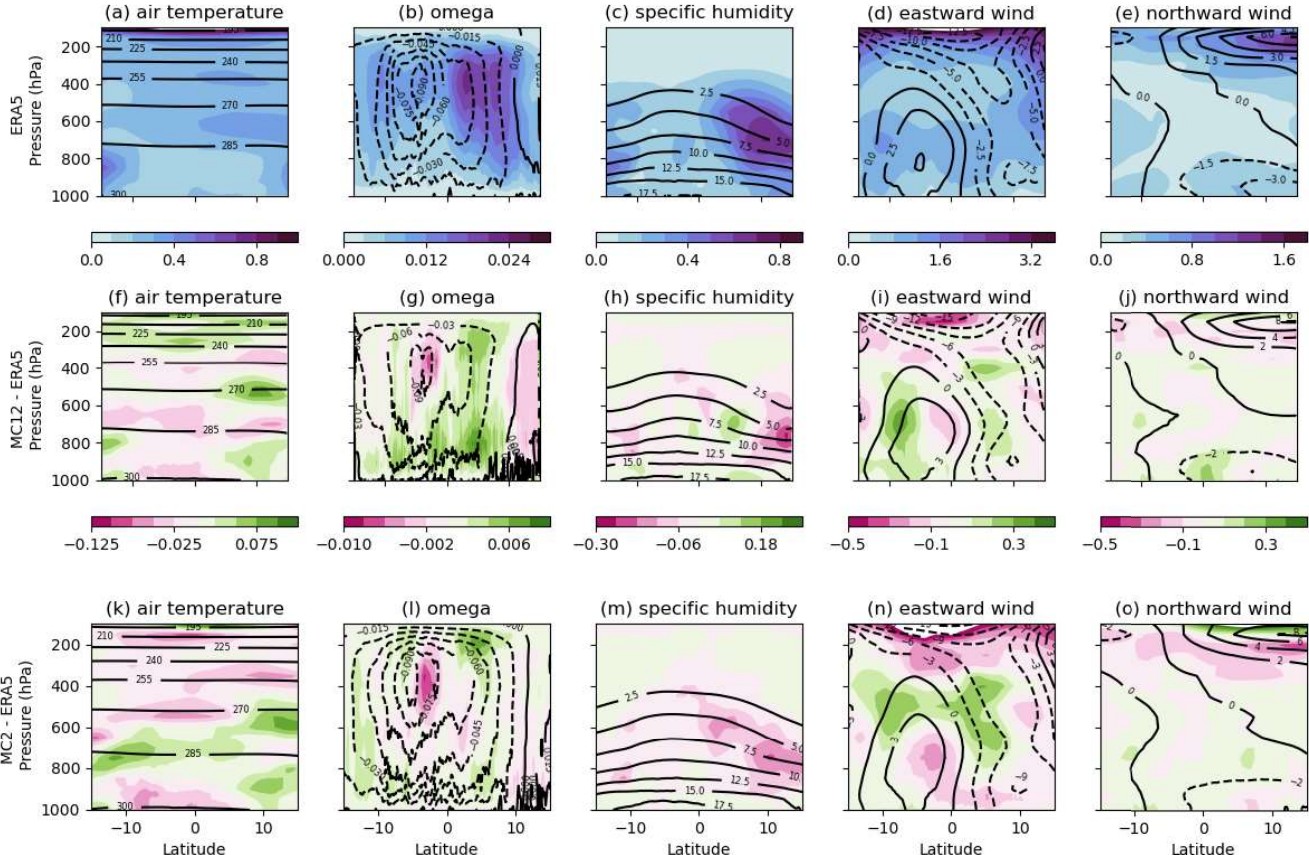

**Figure 11.** Interannual standard deviations of zonal means across the MC2 domain (90° –155° E) of key variables. These standard deviations have been computed by first calculating seasonal means for each of the 10 DJF seasons, and then taking the standard deviation. Top: ERA5, centre: MC2 bias from ERA5, bottom: MC12 bias from ERA5. Filled contours indicate standard deviations while black line contours indicate the mean states. Variables are as labelled.

rainfall for the simulated 10 seasons. The large degree of variability in the northern extent of the equatorial rainband is clear from this figure. Furthermore, MC12 and MC2 replicate this quite well, despite their overall over-prediction of the magnitude of seasonal rainfall. 2016-17 (grey broken line) and 2017-18 (olive solid line) show the most northerly extent of the rain-band consistently across models and observations, while El Niño seasons 2015-16 (pink dotted line) and 2009-10 (red broken line) show the least. Differences between the models and observations are still present, however, for example 2018-19 (light blue dashed line) is much drier than 2015-16 (pink dotted line) in MC12 but not in the observations. In both models, the Pearson correlation coefficient over year and latitude of the data presented in panels (a) and (b) with the data presented in panel (c), subsetted to the MC2 domain, was 0.94.

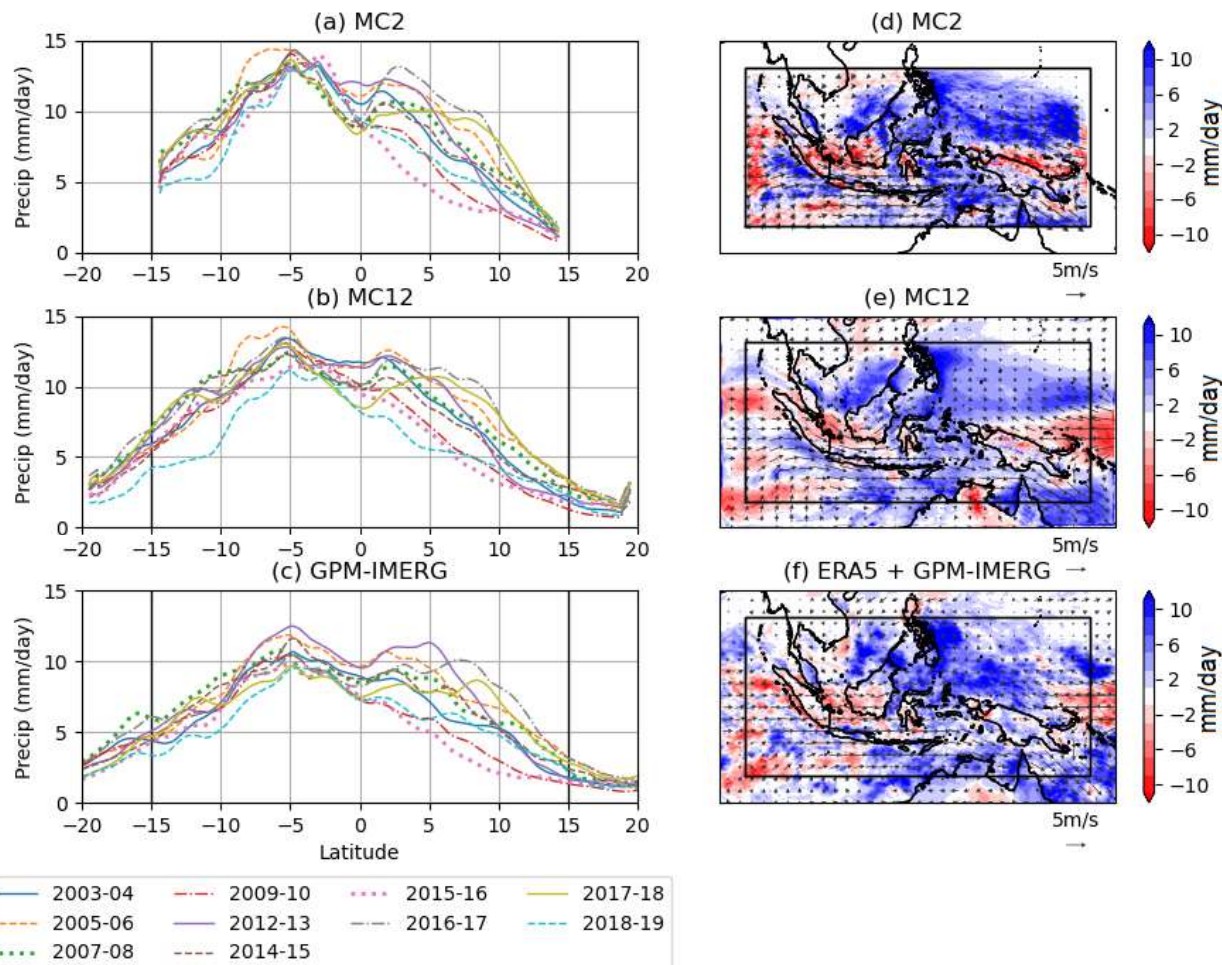

**Figure 12.** Left: precipitation totals by latitude in each season. Colours indicate the different seasons. Right: 850 wind and precipitation differences between 2007-08 (La Niña) and 2015-16 (El Niño). Top: MC2, centre: MC12, bottom: ERA5. TODO:correlations

Further examination of representation of ENSO is shown in the right panels of Figure 12. These figures show maps of the differences in 850 hPa winds and precipitation between 2007-08 and 2015-16. These years have been selected as the most intense La Niña and El Niño in the sample. Large-scale features are common across the models and observations: The La Niña season featured more rainfall over the oceans, particularly to the east of 120° E. The El Niño season features more rainfall in Sumatra, southern Borneo and the western Java Sea. Strong relative easterlies in the far east of the domain during the La Niña Season are associated with relative drying compared with La Niña, while the equatorial westerlies are weakened in the El Niño season. A key difference between models and observations is that both models show a westward extension of the location of higher El Niño rainfall and the corresponding easterly strengthening in the winds in the east of the domain, stretching to the north coast of New Guinea. By contrast in observations, this feature does not extend west of the Solomon Islands.

It must be stressed that this analysis was conducted only for the developing phase of each ENSO year, corresponding to the simulation period (DJF). This matters because ENSO-induced rainfall anomalies over the MC vary greatly both seasonally and regionally. This is a general limitation of this dataset for analysis of seasonal and longer-timescale variability, and their associated teleconnections.

## 5  Air-Sea coupling

This final section studies aspects of the air-sea coupling between the MetUM atmosphere and the KPP-ocean. The purpose of the simulations described in this paper is to provide a framework for investigating the important convective and convectively-coupled processes over the Maritime Continent, and how they are represented in models. It is well-known that important differences arise in the relationship between convection and SST on intraseasonal timescales between coupled and atmosphere-only simulations (see e.g. Figure 7 of Kim et al., 2010). In atmosphere-only models on intraseasonal timescales convection tends to become in phase with SST as a result of the higher boundary layer moist static energy, whereas in coupled simulations the high SSTs are associated with periods of clear skies and low windstresses which lead to ocean warming. Conversely, SST is negatively correlated with earlier precipitation due to associated cloudiness and the injection of cool fresh rainwater at the surface (Kumar et al., 2013). In the MJO this leads to a quadrature between convection and SST in observations and coupled models compared to a more in-phase relationship between convection and SST in atmosphere only models.

Ideally to examine the role of air-sea coupling, a corresponding set of atmosphere-only simulations could be examined as a baseline. However, due to the computational expense of the convection-permitting simulations, this was not feasible. Instead, a single season (2015-16) has been run in atmosphere only model, using 3-day updating OSTIA SSTs as the lower boundary conditions. With a single season we cannot demonstrate robustly whether the mean-state and variability of the coupled runs is more or less realistic than for atmosphere only runs, but the purpose of this paper is to demonstrate that the coupled runs produce a realistic mean state and variability and are thus suitable for process studies where air-sea interaction is important.

Figure 13 shows the lead-lag correlations between daily precipitation and sea surface temperatures across the two models and in a combination of observational product (OSTIA SST and GPM-IMERG precipitation). Correlations are calculated on the MC12 grid, and then spatially averaged across the inner domain. Latitudes north of $3°$ N are excluded due to low rainrates. As per Kumar et al. (2013), the correlation switches from positive to negative near lag 0 in all three datasets (Figure 13). MC2 shows good performance compared to observations, while MC12 has a higher amplitude than the observational product or MC2. This result suggests that precipitation in MC12 is overly sensitive to SST, and that this issue is resolved in MC2. Both modelled products peak at lead 4. The observational correlation has lower amplitude extrema, peaking near $\pm 0.1$ at lead and lag 6. However, we note that observational uncertainty, any smoothing, and the independent methodologies used in the production of OSTIA and GPM-IMERG may result in reduced correlations for the observational correlation compared to a 'true' atmosphere.

To demonstrate the robust qualitative difference in the phase relationship, the right panel of Figure 13 shows the grid-point lead-lag relationship between precipitation and SST for the 2015-16 season atmosphere-only runs, compared to the same season

for the coupled runs. The correlations are weaker at all times in both MC2 and MC12 for the atmosphere-only run. Moreover, the shape of the lead-lag relationship is different: SST and precipitation are approximately in quadrature in the coupled run, whereas the atmosphere-only runs are much more in-phase, with positive correlation at lead/lag 0, compared to approximately zero correlation in the coupled runs. We note that in this configuration, with both the boundary conditions and the prescribed SSTs originating from observations, the relationship between SST and convection is perhaps more constrained to be close to the observed relationship than in a free running simulation, particularly on intraseasonal timescales, but even within that constraint the differences between the coupled and atmosphere-only simulations are large.

Figure 14 shows the precipitation anomalies for this single season in the uncoupled and coupled simulations. The persistent dry biases to the south of New Guinea, Java and Sumatra in the atmosphere-only simulations are resolved in the coupled models. Wet biases to the north of PNG are slightly intensified. In the convection parametrised simulations, a wet bias is present in the South China Sea in both the coupled and atmosphere only model. In general, although the magnitude of the biases is comparable across the simulations, the coupled MC2 simulation shows a greater mix of wet and dry biases, and smaller regions of persistent biases with the same sign. This finding suggests that the coupled MC2 biases are derived to a greater degree by internal variability in this 3-month simulation, which averages out in longer simulations as the signal to noise ratios increase.

The results shown in Figure 4 support this interpretation: averaged over all seasons, the coupled MC2 simulation shows a consistent 1-3 mm/day oceanic wet bias away from the domain boundaries, comparing favourably to previous atmosphere-only convection-permitting simulations, which show 4-7 mm/day oceanic dry biases (e.g. Vincent and Lane, 2017).

This is a long-standing problem in simulations over the Maritime Continent: atmosphere-only models tend to underestimate mean precipitation over the MC (e.g. Neale and Slingo, 2003; Toh et al., 2018), while coupled models tend to overestimate it (e.g. Inness and Slingo, 2003; Liu et al., 2023).

The propagation of the MJO through the mixed-layer ocean is examined in Figure 15. Consistent with Figure 9, the real-world RMM index is used to construct phase composites. Both MC2 and MC12 show largely similar propagation behaviour. In each case, high diurnal SST amplitudes, expected to be associated with clear conditions, are followed by increases to the foundational SSTs. These in turn lead to high precipitation and the active phase of the MJO, which is followed by a deepening of the oceanic mixed layer, reduced sea surface tmperatures and a reduced diurnal cycle of SST completing the cycle.

## 6    Discussion and Conclusions

As a part of the TerraMaris project, 10 DJF seasons of continuous coupled atmosphere-ocean model runs in the Maritime Continent region have been simulated at a convection-permitting resolution by the MetUM coupled to the KPP model. These simulations add to a growing collection of multi-season regional convection-permitting climate simulations (e.g. Stratton et al., 2018; Vincent and Lane, 2018). Our simulations represent important computational advances through the use of a large domain covering all of the Maritime Continent region at 2-km grid-spacing, and through coupling with a mixed-layer ocean model. In the Maritime Continent, where the moisture and energy driving the region's intense tropical convection is sourced from warm,

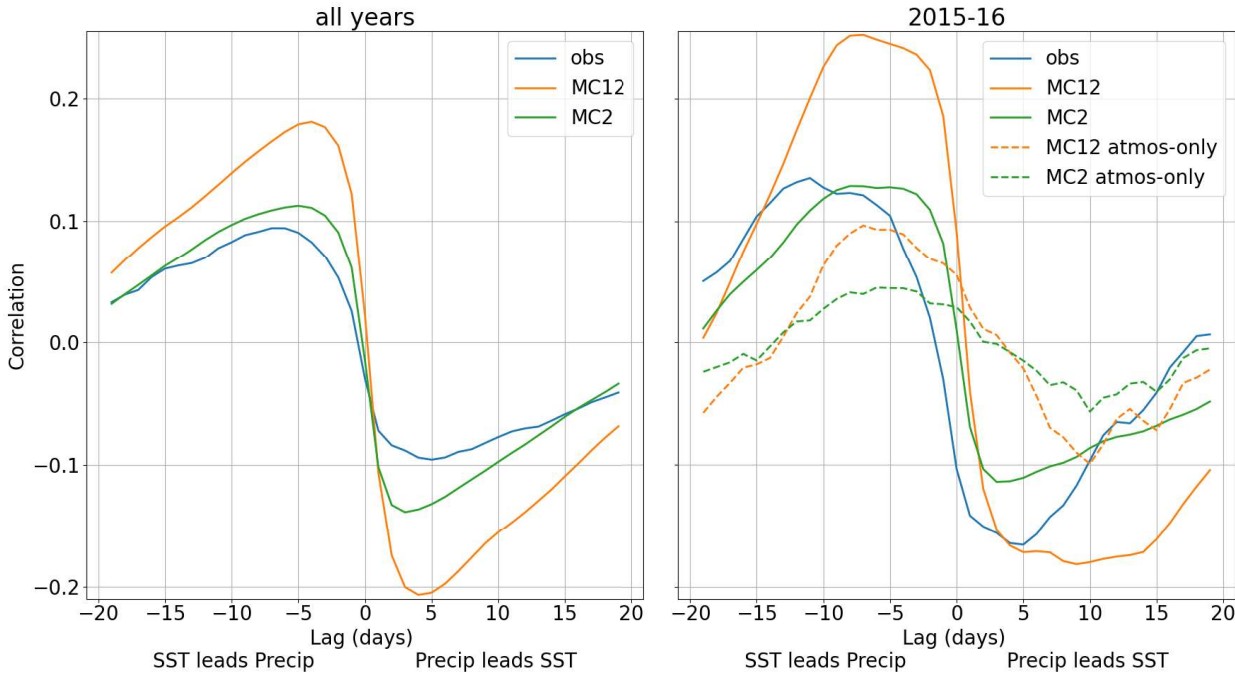

**Figure 13.** Grid-point lead-lag relationship between precipitation and SST averaged across ocean grid-cells between 15° S and 3 ° N. Each season was linearly detrended before computation to remove seasonal and interannual variability. Coloured lines indicate observations in blue (OSTIA compared to GPM-IMERG), MC12 in orange and MC2 in green. Panel a) shows the lead-lag relationship for both coupled simulations suites and observations for all simulation years; panel b) shows the lead-lag relationship for the coupled (solid lines) versus atmosphere-only (dashed lines) simulations for the 2015-16 season only.

shallow seas, mixed-layer coupling allows for the inclusion of a crucial component of the Maritime Continent's climate. The design of the KPP-ocean avoids the introduction of large SST biasesthat are often seen in coupled models of the Maritime Continent (Wang et al., 2022; Xue et al., 2020). As a result of KPP-coupling, a realistic lead-lag relationship between precipitation and SST is simulated. This simple mixed-layer coupling methodology shows promise for convection-permitting simulations across the tropics.

This paper has compared 10 winter seasons of convection-permitting coupled regional seasonal simulations of the MC region with a corresponding parametrised convection model. The mixed layer coupling and ocean dynamics parametrisation in the KPP model component has ensured that SST biases remain low throughout the simulations and results in reduced oceanic dry biases compared to uncoupled simulations at both resolutions, but often overcompensates to generate wet biases in MC12. This partial improvement to precipitation bias fits well with the findings of the papers summarised by Xue et al. (2020).

This paper has demonstrated that the convection-permitting MC2 and the convection parametrised MC12 simulations are capable of representing the vertical structure of the atmosphere, eastward and westward-propagating signals chiefly due to

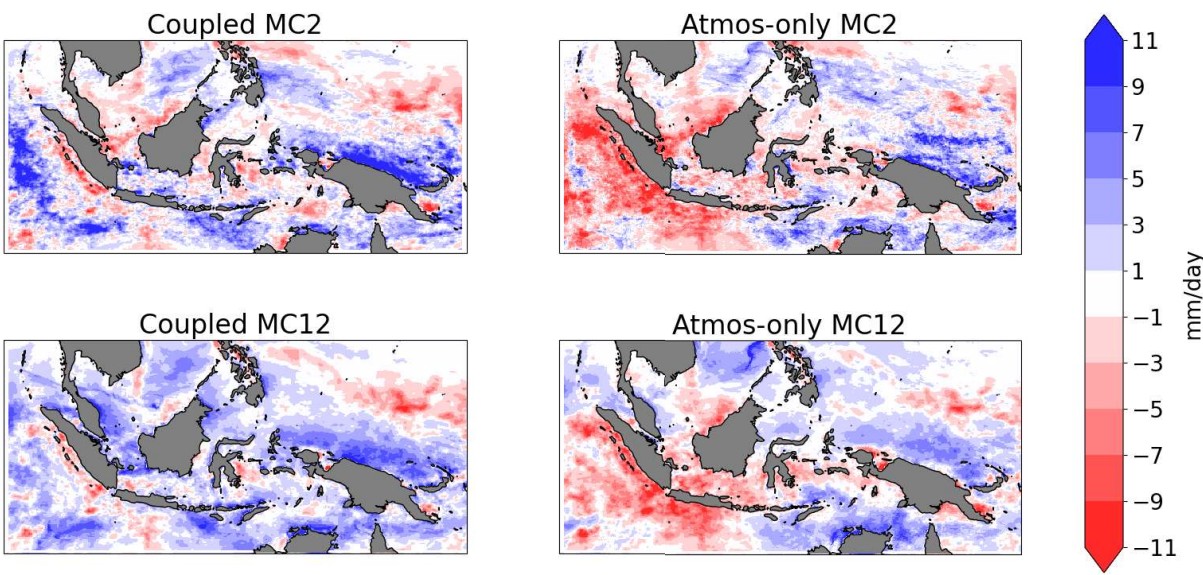

**Figure 14.** Oceanic precipitation biases for the 2015-16 season, relative to GPM-IMERG. Top: MC2, bottom: MC12. Left: coupled, right: uncoupled.

equatorial waves, and the interannual variability of low-level winds and precipitation due to modes of variability such as ENSO. Improvements to the diurnal cycle of precipitation such as those seen here are widespread across convection-permitting simulations (e.g., Birch et al., 2015; Stratton et al., 2018; Vincent and Lane, 2018). The MJO is present, but is largely forced
at the boundaries and generally propagates too quickly. Consistent with the findings of Birch et al. (2015), the magnitude of the MJO precipitation signal over land in MC12 is small and is larger in MC2, both compared to GPM-IMERG. Additionally, MC2 accurately simulates the diurnal cycle of convection over land and coastal oceans, which suggests it may be simulating diurnally propagating convection.

At large length-scales and long time scales, both models were found to be strongly coupled to the driving ERA5 boundary
conditions (Figures 7, 8). This finding has several implications for the future research applications of the datasets. For example, tight coupling ensures that the MJO will be present in the models, even if the models would struggle to independently generate it. This enables further studies such as of the dependence of convection on the MJO phase. Furthermore, it allows for case-study comparisons and the straightforward application of real-world MJO indices for compositing. However, tight control of the large-scale by the boundary conditions limits the degree to which large-scale flow is able to respond to convection. This
means that studies of scale-interactions between convection and the large-scale will need to be carefully planned and may be required to focus on smaller features such as equatorial waves.

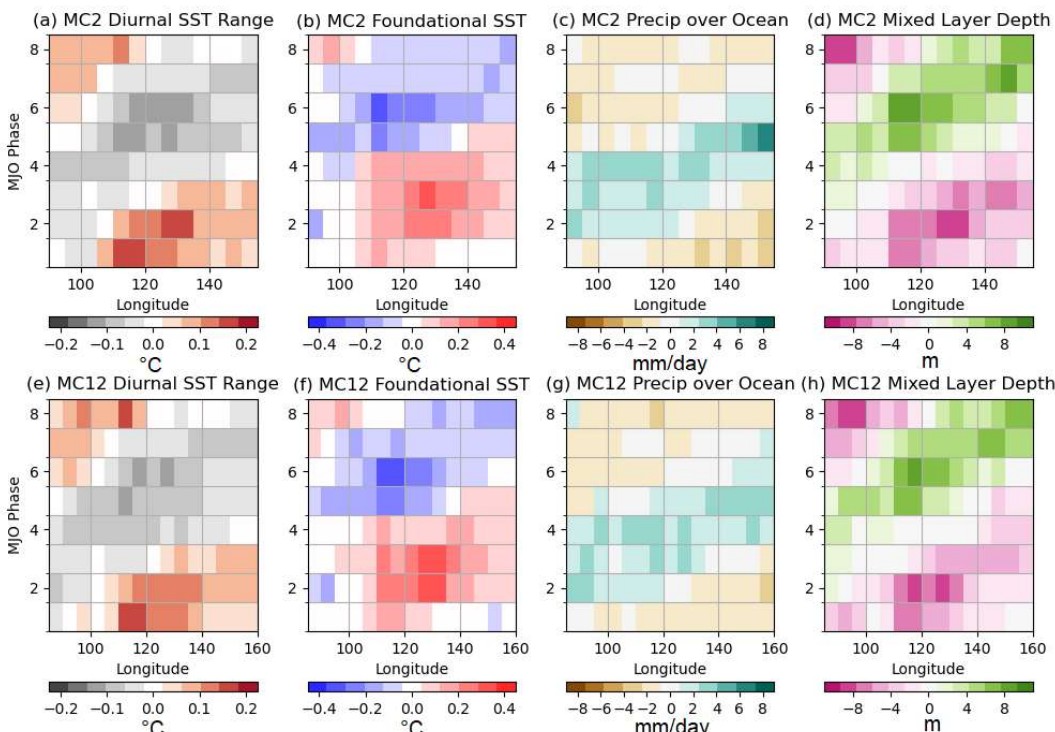

**Figure 15.** KPP-ocean anomaly parameters by longitude and MJO phase. KPP-ocean anomalies are averaged across 15S-2N and discretised to 5° longitude bands. From left to right: Diurnal SST amplitude, foundational SST, precipitation, mixed layer depth. Top: MC2, bottom: MC12.

The KPP mixed layer ocean model was included in these simulations in order to better simulate the thermal coupling of the ocean to the atmosphere. As this model does not simulate dynamical processes, the resultant ocean model output is oversimplified in many ways and is not representative of true oceanic variability. This approach side-steps the issue of poor representation of the Indonesian through-flow highlighted by Xue et al. (2020). These simulations may be considered to be atmospheric model simulations with an improved lower boundary condition, rather than fully coupled climate simulations. Thus the evaluation of this paper has focussed largely on atmospheric processes. Further investigation of the KPP-ocean may yield interesting insights if carefully framed, for example through a 'mechanism denial' lens in comparison to a full ocean model.

Future analysis of these simulations will focus on at least three areas. The first will consider the contributions of different classes of convective organisation to the variability of convective heating over the MC region under the two convection frameworks (permitting and parametrised), following the approach of Holloway et al. (2013). The second stream will study the representation of diurnally propagating convection across the two models. Further work will also further examine the value of the air-sea coupling in convective-permitting simulations through more comprehensive comparisons with uncoupled models.

This modelling framework also has the potential to act as a test-bed for the evaluation of future convection parametrisation schemes through updated MC12 simulations which can be compared against the existing MC2 model output presented here.

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

*Author contributions.* SJW and NK designed the experiment with input from CB, and AM. CS and NK developed the coupled model configuration. EH set up and performed the simulations. SJW and EH designed the analysis with input from CB, AM, and SP. EH performed the analysis and wrote the paper. DS assisted with the analysis of the diurnal cycle in GPM-IMERG. SW, CB, EH, DS and SP edited the manuscript.DS led the response to reviewers, including additional analysis, with assistance from SJW and EH.

*Competing interests.* The contact author has declared that none of the authors has any competing interests

*Acknowledgements.* This work was funded by the U.K. National Environmental Research Council (NERC) TerraMaris project (NE/R016739/1).SJW
was also supported by the National Centre for Atmospheric Science including through the NERC National Capability International Pro-
grammes Award (NE/X006263/1) . This work was undertaken with the assistance of resources and services from the ARCHER2 and JASMIN
computing resources. Computational support was provided by the NCAS CMS team. Valuable and constructive comments by C. Stassen and
B. White are gratefully acknowledged.Thanks to two anonymous reviewers for comments that greatly improved the manuscript.