# Peer review of "Evaluation of multi-season convection permitting atmosphere mixed layer ocean simulations of the Maritime Continent"

_Geoscientific Model Development, 2023_

## Referee Comment (RC1)

**Reviewer's report for manuscript:**

**"Evaluation of multi-season convection permitting atmosphere - mixed layer ocean simulations of the Maritime Continent".**

**General comments**

This paper presents a thorough intercomparisons between the outputs of convective permitting regional climate model and convection parameterised simulation, encompassing various aspects such as: the mean state of precipitation, its diurnal cycle, SST, the representation of equatorial waves and the teleconnection of precipitation with different modes of variability (MJO, ENSO). The evaluations employ both statistic-based metrics and process/regime-based metrics, contributing credibility and bring novelty to this research. The manuscript is well-written with a narrative and easy-to-follow story. Therefore, a "minor revision" is recommened for this current manuscript. Detailed comments for this minor revision are provided below with the hope that they will help to improve the paper's quality.

**Comments**

1. In general, both of models exhibit a wet bias compared to GMP-IMERGE. Interestingly, more intense precipitation in convective-permitting model compared with the convection parameterised model. It would be beneficial to explicitly explain the reason for these wetter biases. Is there any potential link to the impact of higher resolution on model biases?

2. Please specific the version of GPM-IMERGE used (satellite with or without correction to rain gauge?). Given large observational uncertainties in estimating precipitation over the Maritime Continent due to lack of station network, it is advantageous if the author can conduct additional analysis on multiple observational precipitation products. Some sub-daily products from different sources (in situ, satellite with correction to rain gauge) could be suggested including: GSDR (A global sub-daily rainfall dataset, Lewis et al. 2019); MSWEP (Multi-source weighted-ensemble precipitation, Beck et al 2017). This additional analysis will bring more insights into how observational uncertainties may impact the model evaluation.

3. Regarding Figure 12: was this analysis conducted during the developing phase (DJF) of the ENSO year? Over Maritime Continent, the observed teleconnections vary largely regionally and seasonally. In general, the ENSO-induced summer rainfall variability is dominant over most part of Maritime Continent (e.g., Sumatra, Java Island), while the spring variability is northern Borneo and Philippines (Wang et al., 2020; Chen et al.,

2023). Given this lead-lag teleconnection, it would be preferable to conduct this analysis over MJJASO of the developing year.

---

## Referee Comment (RC2)

This manuscript presents a comprehensive evaluation of a multi-season convection-permitting regional climate simulation for the Maritime Continent, using the Met Office Unified Model with a fine 2.2-km grid spacing. This work incorporates atmosphere-ocean coupling with the K profile parametrization mixed layer ocean model. The study contrasts this simulation with a convection-parametrized model, deriving insights from ERA5 reanalysis. The paper outlines the configuration and performance of both models, noting minor biases in sea surface temperature and precipitation. The convection-permitting model shows improvements in the diurnal cycle and equatorial waves representation. The paper is generally well written, organized and analyzed. I recommend publication after the authors have addressed the comments below.

General comments:

1. The authors emphasized the importance of the air-sea interaction. However, the air-sea coupling is just tested for one single season. I wonder whether the testing is reliable. Does the result depend on the case/season you chose?

2. In this paper, the biases are generally shown in the distribution plots except for Figure 3g. It is better if the comparisons between the two models can be quantitate. I recommend the authors giving the root mean square error of each model compared to the reference dataset for each variable in Figures 4-6 and 9-12.

3. Air temperature and cloud fractions are also important variables for climate analysis. How do the performances for both mean states and variability of these two variables?

4. The writing style leans toward descriptiveness and narrative. While this is not inherently wrong or inappropriate. I believe that the GMD journal also encourages further exploration and digging beyond the mere description of data results. For example, the MC2 SST biases are generally weaker than the MC12 (Figures 3 and 10). In contrast, the wet bias over the ocean seems be stronger in MC2 than MC12 (Figures 4 and 10). How do the changes in resolution impact SST and precipitation?

---

## Author Comment (AC1)

**TerraMaris model evaluation paper: response to reviewers**

Emma Howard et al.

February 8, 2024

**Response to Reviewer 1**

We thank the reviewer for their positive review of the manuscript. Their specific comments will improve the paper, especially the discussion of precipitation in our two simulation suites.

**Replies to specific comments**

1. A wet bias over the Maritime Continent is common for atmosphere-ocean coupled models (Roberts et al., 2019), especially in DJF (Liu et al., 2023). This is a long-standing problem in simulations over the Maritime Continent: atmosphere-only models tend to underestimate mean precipitation over the MC (e.g. Neale and Slingo, 2003; Toh et al., 2018), while coupled models tend to overestimate it (e.g. Inness and Slingo, 2003; Liu et al., 2023).

   It is also common for convection-permitting models to exacerbate wet or dry biases compared to the same model run with parametrized convection; this is certainly true for the Met Office Unified Model (e.g. Muetzelfeldt et al., 2021), and in general there is no systematic improvement in mean precipitation bias for convection-permitting models over parametrized models (Prein et al., 2015). That the strongest biases in MC2 are close to the inflow boundaries is also consistent with other regional convection-permitting modelling over the MC (Jones et al., 2023) [note that they only see the strong wet bias close to the eastern boundary, but this is because their analysed period 1-11th Jan 2018 had a consistent strong easterly over that part of the domain, whereas the winds at the western boundary fluctuated much more]. This is likely due to differences in model physics between the parametrized and explicit versions of the model; for instance, if the explicit physics prefers a slightly drier atmosphere (consistent with manuscript Fig. 6b), then excess moisture advected in at the boundaries will be rapidly precipitated out.

   There is certainly an impact of higher resolution on model biases, but this link is complex (e.g. Roberts et al., 2019; Holloway et al., 2013). It is not possible to perform a clean analysis of the effects of model resolution on precipitation bias using this dataset, due to the confounding effects of very different model physics — especially parametrized versus explicit representations of convection.

   We shall add text to this effect to the discussion of Figure 4 in the manuscript.

2. All analyses of precipitation used GPM-IMERG v06B. We used the "precipitationCal" product, which is satellite observations including correction to rain gauge.

   In response to the reviewer's comment regarding observational uncertainty, we have reproduced Figure 4 of the manuscript using the "past" precipitation product from MSWEP (Beck et al., 2017) (see attached Figure 1). Since the rain gauge network is very sparse over the Maritime Continent, the algorithms used by both GPM-IMERG and MSWEP weight the satellite observations much more highly than in regions with dense rain gauge coverage, so differences in the datasets over the Maritime Continent should largely be due to different satellite retrieval methods. The differences between the two simulation runs, and the differences between the runs and either precipitation dataset, are much larger than the difference between the two individual datasets. In particular, the spatial patterns are very similar for both satellite datasets.

   We therefore conclude that the rain gauge-corrected satellite observational uncertainty is less important for the mean precipitation than the difference between the two models, and than the difference between the models and those observations.

   For the interannual variability, GPM has a fairly uniformly positive interannual standard deviation anomaly over ocean, and close to zero anomaly over land, relative to MSWEP (see attached Figure 2). This strengthens the conclusion that MC2 overestimates the interannual precipitation variability everywhere. For MC12, the interannual standard deviation of precipitation over ocean is generally weakly overestimated relative to MSWEP, compared to underestimated relative to GPM. For both simulations,

the relative spatial uniformity of the GPM minus MSWEP anomaly causes the spatial pattern of the model biases to be broadly similar relative to both reference datasets.

In previous work, differences in diurnal cycle phase have been found to be negligible between different precipitation products over the Maritime Continent, especially at the 3 hour temporal sampling rate of many observational datasets (see e.g. Supplementary Figure 7 of Dong et al., 2023).

We shall add Figures 1 and 2 as supplementary material to the revised manuscript, along with a brief summary of the above explanatory text.

It is an excellent point that rain gauge coverage of the Maritime Continent is very poor (see e.g. Figs. 2, 3, B4, B8 of Lewis et al., 2019). This is especially true over areas with high orography, where the MC2 and MC12 biases are worst compared to GPM-IMERG and MSWEP, and of course over the ocean, where there is a general wet bias in both models. Given this fact, we feel that it is beyond the scope of this paper to perform a direct comparison with gauge data. We will instead add a sentence regarding the lack of rain gauge data in our discussion of the precipitation analysis.

3. All of our simulations were run for Boreal winter seasons (NDJF); analyses were conducted for DJF, with November discarded for spin-up. The reasoning for this is that DJF is the most active period for intraseasonal variability over the Maritime Continent (explained in line 87 of the manuscript). Therefore, while it would be an interesting research avenue, it is beyond the scope of this paper to re-run the two simulation suites for MJJASO. We will acknowledge this limitation explicitly in the "Discussion and conclusions" section of the revised manuscript.

[Figure]

Figure 1: Mean precipitation and biases (mm day$^{-1}$). Subplots along the diagonal indicate MC2, MC12, GPM-IMERG, and MSWEP respectively. Upper off diagonal subplots show difference plots between each of the datasets.

[Figure]

Figure 2: Interannual standard deviation of precipitation and biases (mm day$^{-1}$). Subplots along the diagonal indicate MC2, MC12, GPM-IMERG, and MSWEP respectively. Upper off diagonal subplots show difference plots between each of the datasets.

**References**

Beck, H. E., van Dijk, A. I. J. M., Levizzani, V., Schellekens, J., Miralles, D. G., Martens, B., and de Roo, A.: MSWEP: 3-hourly 0.25° global gridded precipitation (1979–2015) by merging gauge, satellite, and reanalysis data, Hydrology and Earth System Sciences, 21, 589–615, https://doi.org/10.5194/hess-21-589-2017, 2017.

Dong, W., Krasting, J. P., and Guo, H.: Analysis of Precipitation Diurnal Cycle and Variance in Multiple Observations, CMIP6 Models, and a Series of GFDL-AM4.0 Simulations, Journal of Climate, 36, 8637 – 8655, https://doi.org/10.1175/JCLI-D-23-0268.1, 2023.

Holloway, C. E., Woolnough, S. J., and Lister, G. M.: The effects of explicit versus parameterized convection on the MJO in a large-domain high-resolution tropical case study. Part I: Characterization of large-scale organization and propagation, Journal of the Atmospheric Sciences, 70, 1342–1369, https://doi.org/https://doi.org/10.1175/JAS-D-12-0227.1, 2013.

Inness, P. M. and Slingo, J. M.: Simulation of the Madden–Julian Oscillation in a Coupled General Circulation Model. Part I: Comparison with Observations and an Atmosphere-Only GCM, Journal of Climate, 16, 345 – 364, https://doi.org/10.1175/1520-0442(2003)016¡0345:SOTMJO¿2.0.CO;2, 2003.

Jones, R. W., Sanchez, C., Lewis, H., Warner, J., Webster, S., and Macholl, J.: Impact of Domain Size on Tropical Precipitation Within Explicit Convection Simulations, Geophysical Research Letters, 50, e2023GL104 672, https://doi.org/https://doi.org/10.1029/2023GL104672, e2023GL104672 2023GL104672, 2023.

Lewis, E., Fowler, H., Alexander, L., Dunn, R., McClean, F., Barbero, R., Guerreiro, S., Li, X.-F., and Blenkinsop, S.: GSDR: A Global Sub-Daily Rainfall Dataset, Journal of Climate, 32, 4715 – 4729, https://doi.org/10.1175/JCLI-D-18-0143.1, 2019.

Liu, S., Raghavan, S. V., Ona, B. J., and Nguyen, N. S.: Bias evaluation in rainfall over Southeast Asia in CMIP6 models, Journal of Hydrology, 621, 129 593, https://doi.org/https://doi.org/10.1016/j.jhydrol.2023.129593, 2023.

Muetzelfeldt, M. R., Schiemann, R., Turner, A. G., Klingaman, N. P., Vidale, P. L., and Roberts, M. J.: Evaluation of Asian summer precipitation in different configurations of a high-resolution general circulation model in a range of decision-relevant spatial scales, Hydrology and Earth System Sciences, 25, 6381–6405, https://doi.org/10.5194/hess-25-6381-2021, 2021.

Neale, R. and Slingo, J.: The Maritime Continent and its role in the global climate: A GCM study, Journal of Climate, 16, 834–848, https://doi.org/10.1175/1520-0442(2003)016¡0834:TMCAIR¿2.0.CO;2, 2003.

Prein, A. F., Langhans, W., Fosser, G., Ferrone, A., Ban, N., Goergen, K., Keller, M., Tölle, M., Gutjahr, O., Feser, F., Brisson, E., Kollet, S., Schmidli, J., van Lipzig, N. P. M., and Leung, R.: A review on regional convection-permitting climate modeling: Demonstrations, prospects, and challenges, Reviews of Geophysics, 53, 323–361, https://doi.org/https://doi.org/10.1002/2014RG000475, 2015.

Roberts, M. J., Baker, A., Blockley, E. W., Calvert, D., Coward, A., Hewitt, H. T., Jackson, L. C., Kuhlbrodt, T., Mathiot, P., Roberts, C. D., Schiemann, R., Seddon, J., Vanniére, B., and Vidale, P. L.: Description of the resolution hierarchy of the global coupled HadGEM3-GC3.1 model as used in CMIP6 HighResMIP experiments, Geosci. Model Dev., p. 4999–5028, https://doi.org/10.5194/gmd-12-4999-2019, 2019.

Toh, Y. Y., Turner, A. G., Johnson, S. J., and Holloway, C. E.: Maritime Continent seasonal climate biases in AMIP experiments of the CMIP5 multimodel ensemble, Climate Dynamics, 50, 777–800, https://doi.org/10.1007/s00382-017-3641-x, 2018.

---

## Author Comment (AC2)

**TerraMaris model evaluation paper: response to reviewers**

Emma Howard et al.

February 8, 2024

**Response to Reviewer 2**

We thank the reviewer for their positive comments on, and constructive criticism of, the manuscript. Their comments will help to improve the quality of the paper, particularly the discussion of air-sea coupling.

**Replies to specific comments**

1. The purpose of the simulations described in this paper is to provide a framework for investigating the important convective and convectively-coupled processes over the Maritime Continent, and how they are represented in models. It is well-known that important differences arise in the relationship between convection and SST on intraseasonal timescales between coupled and atmosphere-only simulations (see e.g. Figure 7 of Kim et al. (2010)). In atmosphere-only models on intraseasonal timescales convection tends to become in phase with SST as a result of the higher boundary layer moist static energy, whereas in coupled simulations the high SSTs are associated with periods of clear skies and low windstresses which lead to ocean warming. In the MJO this leads to an observed quadrature between convection and the SST in observations and coupled models compared to a more in phase relationship between convection and SST in atmosphere only models. We therefore do not intend to demonstrate that the mean-state and variability produced by the coupled runs are *better* than the same atmospheric model runs with prescribed SSTs when compared to observations; rather, we intend to demonstrate that coupled runs produce realistic mean-state and variability.

   We will re-word Section 5 on Air-Sea coupling to clarify this aspect.

   To demonstrate the robust qualitative difference in the phase relationship, we will add a second panel to Figure 13, showing the grid-point lead-lag relationship between precipitation and SST for the 2015-16 season atmosphere-only runs, compared to the same season for the coupled runs (see attached Figure 1). The correlations are weaker at all times in both MC2 and MC12 for the atmosphere-only run. Perhaps more importantly, the shape of the lead-lag relationship is different, with positive correlation at lead/lag 0, compared to approximately zero correlation in the coupled runs. This is in line with other studies in atmosphere-only models (with a fairly symmetric relationship about lag 0 since the SSTs cannot respond to the precipitation) and coupled runs (where higher SSTs promote the development of higher rainfall, but the precipitation and associated increased cloud cover then cool the SSTs). We note that in this configuration with both the boundary conditions and the prescribed SSTs coming from observations the relationship between SST and convection is perhaps more constrained to be close to the observed relationship than in a free running simulation, but even within that constraint the differences between the coupled and atmosphere-only simulations are large.

2. We agree that quantifying the overall domain-mean magnitude of the biases is useful. Therefore we will add a table of domain-averaged mean, mean bias, and RMSE (compared to the relevant reference dataset) for each variable considered in Figures 4-6, 9-11. For Figure 12, the mean and RMSE values shall instead be quoted in the accompanying text.

3. We agree that it is important to investigate the representation of clouds when considering convective processes, although cloud fraction itself is not necessarily the most relevant variable for understanding how the variability is represented in models. For example clouds may brighten leading to increases in reflected shortwave radiation or deepen leading to reductions in outgoing longwave radiation without any change in cloud fraction. Furthermore the definition of cloud fraction in models and observations can vary greatly between individual models (or reanalysis products) and depend strongly on the "observation" used to define them and so a direct comparison between models and observations is not always helpful. We have however compared both the outgoing longwave and reflected shortwave from both model simulations against observations (see attached Figures 2, 3) and find that in both MC2 and MC12 simulations the

biases are broadly consistent with typical GCM biases, and that the biases in MC2 tend to be reduced compared to MC12, suggesting a generally better representation of clouds in MC2 (consistent with a higher-resolution convection-permitting simulation). However, as with all other variables considered, the interannual variability is generally too high for both simulations (see attached Figures 4, 5). We will include a short discussion on these biases in the paper; the figures will be included as supplementary material.

We do not agree with the reviewer that biases in the 2m air temperature are particularly important for the kind of process studies for which we expect these simulations to be used. Moreover, over land, surface air-temperature will be particularly sensitive to orographic height and therefore not easy to compare even between the two models where the orography is different. We have therefore not considered this further.

4. We agree that the style of the paper is largely descriptive. We will add a few more explanatory/exploratory comments, but do not wish to present too much speculation without properly digging in to the causes of differences between the model runs, and between the model runs and observations. The purpose of creating this dataset is to explore precisely these questions of why higher resolution/explicit representation of convection changes e.g. the mean state and variability, including for instance coupling with equatorial waves and the MJO.

   In response to the reviewer's specific queries:

   - On the wet bias being stronger in MC2 than MC12: this was not unexpected, as it is fairly common in convection-permitting simulations (see response to Reviewer 1, Specific Comment 1).

   - On the SST biases being weaker in MC2 than MC12: this was unexpected, as the relaxation runs to determine the KPP flux corrections were only performed for the parametrized convection model at N1280 grid spacing. It would have been entirely reasonable to expect that the SST biases would therefore be worse in MC2.

     A full exploration of the mechanisms behind this difference is beyond the scope of this paper, but a preliminary analysis of the bias in both simulations suggests that the change in SST bias between MC2 and MC12 are broadly consistent with changes in the surface heat flux and these changes are also consistent with changes in the mixed-layer depth. This is especially clear around the islands of the Maritime Continent, with cooler SST and deeper mixed layers associated with lower heat fluxes into the ocean.

     However, the pattern of surface fluxes is complicated. The changes in surface radiation budget are broadly consistent with changes in the corresponding top-of-atmosphere fluxes, with some cancellation between the shortwave and longwave fluxes. The changes in the latent-heat (LH) flux are more complex, noting that the LH flux is defined as positive downwards and as such a positive change indicates decreased evaporation. The sign of the LH flux change is not consistent with it being driven directly by the SST, as the evaporation is generally higher in regions of colder SST. Similarly over large regions away from the islands evaporation is lower, but the surface wind stress is increased. The relative contributions of each component of the surface fluxes varies from region to region.

     To properly understand the differences in the SSTs would require a detailed analysis of the time evolution of the biases and would be regionally sensitive. It would be particularly interesting to look at these processes around the coastlines where it is clear there is a significant change in the representation of the diurnal cycle of convection, which is known to be associated with strong onshore/offshore circulations. This would make a good focus for a future study.

[Figure]

Figure 1: Grid-point Lead-lag relationship between precipitation and SST averaged across ocean grid-cells between 15° S and 3° N for the 2015-16 season, comparing the atmosphere-only and coupled model runs. For each dataset, the season was linearly detrended before computation to remove seasonal variability. Solid lines indicate atmosphere-only runs runs; dashed lines indicate coupled runs; MC2 is shown in green; and MC12 is shown in orange.

[Figure]

Figure 2: Mean outgoing longwave radiation and biases (W m$^{-2}$). Subplots along the diagonal indicate MC2, MC12, and NOAA daily OLR respectively. Upper off diagonal subplots show difference plots between each of the datasets.

top-of-atmosphere outgoing shortwave radiation flux [W m-2]

[Figure]

Figure 3: Mean outgoing shortwave radiation and biases (W m$^{-2}$). Subplots along the diagonal indicate MC2, MC12, and NCEP-NCAR reanalysis respectively. Upper off diagonal subplots show difference plots between each of the datasets.

top-of-atmosphere outgoing longwave radiation flux [W m-2]

[Figure]

Figure 4: Interannual standard deviation of outgoing shortwave radiation and biases (W m$^{-2}$). Subplots along the diagonal indicate MC2, MC12, and NOAA daily OLR respectively. Upper off diagonal subplots show difference plots between each of the datasets.

[Figure]

[Figure]

Figure 5: Interannual standard deviation of outgoing shortwave radiation and biases (W m$^{-2}$). Subplots along the diagonal indicate MC2, MC12, and NCEP-NCAR reanalysis respectively. Upper off diagonal subplots show difference plots between each of the datasets.

[Figure]

Figure 6: Spatial maps of differences between the all-season means of variables linked to the SST. Top row (L–R): sea surface temperature; mixed-layer depth; heat flux into the ocean. Middle row (L–R): downward surface shortwave flux; downward surface longwave flux; downward surface latent heat flux (a positive change corresponds to reduced evaporation). Bottom row (L–R): magnitude of downward surface wind stress; top-of-atmosphere outgoing longwave radiation flux (as an indication of cloud changes); top-of-atmosphere outgoing shortwave radiation flux (as an indication of cloud changes). Differences in downward surface sensible heat fluxes are not shown; their magnitude is less than $\sim 3$ W m$^{-2}$ almost everywhere.

**References**

Kim, H. M., Webster, P. J., Hoyos, C. D., and Kang, I. S.: Ocean–atmosphere coupling and the boreal winter MJO, Climate Dyn., 35, 771–784, https://doi.org/10.1007/s00382-009-0612-x, 2010.

---

## Author Response (AR1)

**Response to Reviewer 1**

We thank the reviewer for their positive review of the manuscript. Their specific comments will improve the paper, especially the discussion of precipitation in our two simulation suites.

**Replies to specific comments**

1. A wet bias over the Maritime Continent is common for atmosphere-ocean coupled models (Roberts et al., 2019), especially in DJF (Liu et al., 2023). This is a long-standing problem in simulations over the Maritime Continent: atmosphere-only models tend to underestimate mean precipitation over the MC (e.g. Neale and Slingo, 2003; Toh et al., 2018), while coupled models tend to overestimate it (e.g. Inness and Slingo, 2003; Liu et al., 2023).

   It is also common for convection-permitting models to exacerbate wet or dry biases compared to the same model run with parametrized convection; this is certainly true for the Met Office Unified Model (e.g. Muetzelfeldt et al., 2021), and in general there is no systematic improvement in mean precipitation bias for convection-permitting models over parametrized models (Prein et al., 2015). That the strongest biases in MC2 are close to the inflow boundaries is also consistent with other regional convection-permitting modelling over the MC (Jones et al., 2023) [note that they only see the strong wet bias close to the eastern boundary, but this is because their analysed period 1-11th Jan 2018 had a consistent strong easterly over that part of the domain, whereas the winds at the western boundary fluctuated much more]. This is likely due to differences in model physics between the parametrized and explicit versions of the model; for instance, if the explicit physics prefers a slightly drier atmosphere (consistent with manuscript Fig. 6b), then excess moisture advected in at the boundaries will be rapidly precipitated out.

   There is certainly an impact of higher resolution on model biases, but this link is complex (e.g. Roberts et al., 2019; Holloway et al., 2013). It is not possible to perform a clean analysis of the effects of model resolution on precipitation bias using this dataset, due to the confounding effects of very different model physics — especially parametrized versus explicit representations of convection.

   We have modified the text in Section 3 in response to this comment (ll. 253–262):

   > Both models exhibit a wet bias compared to the GPM-IMERG climatology averaged over the same period, as indicated by Figure 4. This is consistent with other high resolution coupled models *using the MetUM framework* in the Maritime Continent region (e.g., Roberts et al., 2019), with both parametrised and explicit convection, especially in DJF (Liu et al., 2023). The wet bias over the ocean is stronger in MC12 than MC2 and has a typical magnitude of around 3 mm/day (panels c, e). Both models also have strong wet biases over high orography, and MC12 has a dry bias over lowlands in Sumatra, Borneo and Java. It is common for convection-permitting models to exacerbate wet or dry biases compared to the same model run with parametrised convection; this is certainly true for the Met Office Unified Model (e.g. Muetzelfeldt et al., 2021), and in general there is no systematic improvement in mean precipitation bias for convection-permitting models over -parametrised models (Prein et al., 2015). Strong boundary effects can be seen near the equator in MC2, where localised wet biases are present 3° from the edges of the domain, consistent with other regional convection-permitting modelling over the MC (Jones et al., 2023).

   We also added a sentence to Section 5 on air-sea coupling (ll. 462–464):

   > This is a long-standing problem in simulations over the Maritime Continent: atmosphere-only models tend to underestimate mean precipitation over the MC (e.g. Neale and Slingo, 2003; Toh et al., 2018), while coupled models tend to overestimate it (e.g. Inness and Slingo, 2003; Liu et al., 2023).

2. All analyses of precipitation used GPM-IMERG v06B (clarified in l. 230 of the revised manuscript). We used the "precipitationCal" product, which is satellite observations including correction to rain gauge (clarified in ll. 232–233 of the revised manuscript).

   In response to the reviewer's comment regarding observational uncertainty, we have reproduced Figure 4 of the manuscript using the "past" precipitation product from MSWEP (Beck et al., 2017) (see attached Figure 1). Since the rain gauge network is very sparse over the Maritime Continent, the algorithms used by both GPM-IMERG and MSWEP weight the satellite observations much more highly than in regions with dense rain gauge coverage, so differences in the datasets over the Maritime Continent should largely be due to different satellite retrieval methods. The differences between the two simulation runs, and the

differences between the runs and either precipitation dataset, are much larger than the difference between the two individual datasets. In particular, the spatial patterns are very similar for both satellite datasets.

We therefore conclude that the rain gauge-corrected satellite observational uncertainty is less important for the mean precipitation than the difference between the two models, and than the difference between the models and those observations.

For the interannual variability, GPM has a fairly uniformly positive interannual standard deviation anomaly over ocean, and close to zero anomaly over land, relative to MSWEP (see attached Figure 2). This strengthens the conclusion that MC2 overestimates the interannual precipitation variability everywhere. For MC12, the interannual standard deviation of precipitation over ocean is generally weakly overestimated relative to MSWEP, compared to underestimated relative to GPM. For both simulations, the relative spatial uniformity of the GPM minus MSWEP anomaly causes the spatial pattern of the model biases to be broadly similar relative to both reference datasets.

In previous work, differences in diurnal cycle phase have been found to be negligible between different precipitation products over the Maritime Continent, especially at the 3 hour temporal sampling rate of many observational datasets (see e.g. Supplementary Figure 7 of Dong et al., 2023).

It is an excellent point that rain gauge coverage of the Maritime Continent is very poor (see e.g. Figs. 2, 3, B4, B8 of Lewis et al., 2019). This is especially true over areas with high orography, where the MC2 and MC12 biases are worst compared to GPM-IMERG and MSWEP, and of course over the ocean, where there is a general wet bias in both models. Given this fact, we feel that it is beyond the scope of this paper to perform a direct comparison with gauge data.

We shall add Figures 1 and 2 as supplementary material to the revised manuscript, and summarized the above discussion in Section 3 on the mean state (ll. 268–278):

> Due to the sparsity and incompleteness of rain gauge observations over the MC (e.g. Figs. 2, 3, B4, B8 of Lewis et al., 2019), there is potentially large observational uncertainty in precipitation over our domain of interest. This is especially true over areas with high orography, where the MC2 and MC12 biases are worst compared to GPM-IMERG, and of course over the ocean, where there is a general wet bias in both models. It is therefore beyond the scope of this paper to perform a direct comparison with gauge data. However, to give some insight into how observational uncertainty may impact the model evaluation, we have compared the precipitation climatology of both simulations to the "Past" product from the Multi-Source Weighted-Ensemble Precipitation (MSWEP) v2.8 dataset, which combines satellite and rain gauge observations with reanalysis (Beck et al., 2019). Mean biases relative to MSWEP were similar both in magnitude and spatial pattern to those relative to GPM-IMERG (Supplementary Figure S02). Differences in diurnal cycle phase have been found to be negligible between different precipitation products over the Maritime Continent, especially at the 3 hour temporal sampling rate of many observational datasets (see e.g. Supplementary Figure 7 of Dong et al., 2023).

and in Section 4.2 on interannual variability (ll. 377–381):

> Given the uncertainty in precipitation observations, we again compared the model interannual standard deviations of precipitation to MSWEP (Supplementary Figure S7). GPM-IMERG has notably more interannual variability over ocean than MSWEP, but the variability over land is comparable, likely due to the rain-gauge correction of both datasets. This strengthens the conclusion that MC2 overestimates the interannual variability of precipitation, but weakens confidence in the sign of the bias over ocean for MC12.

3. All of our simulations were run for Boreal winter seasons (NDJF); analyses were conducted for DJF, with November discarded for spin-up. The reasoning for this is that DJF is the most active period for intraseasonal variability over the Maritime Continent (explained in line 87 of the manuscript). Therefore, while it would be an interesting research avenue, it is beyond the scope of this paper to re-run the two simulation suites for MJJASO. We have acknowledged this limitation explicitly in the "Discussion and conclusions" section of the revised manuscript (ll 409–412):

> It must be stressed that this analysis was conducted only for the developing phase of each ENSO year, corresponding to the simulation period (DJF). This matters because ENSO-induced rainfall anomalies over the MC vary greatly both seasonally and regionally. This is a general limitation of this dataset for analysis of seasonal and longer-timescale variability, and their associated teleconnections.

[Figure]

Figure 1: Mean precipitation and biases (mm day$^{-1}$). Subplots along the diagonal indicate MC2, MC12, GPM-IMERG, and MSWEP respectively. Upper off diagonal subplots show difference plots between each of the datasets. This figure appears as Supplementary Figure S2 in the revised manuscript.

[Figure]

Figure 2: Interannual standard deviation of precipitation and biases (mm day$^{-1}$). Subplots along the diagonal indicate MC2, MC12, GPM-IMERG, and MSWEP respectively. Upper off diagonal subplots show difference plots between each of the datasets. This figure appears as Supplementary Figure S7 in the revised manuscript.

**Response to Reviewer 2**

We thank the reviewer for their positive comments on, and constructive criticism of, the manuscript. Their comments will help to improve the quality of the paper, particularly the discussion of air-sea coupling.

**Replies to specific comments**

1. The purpose of the simulations described in this paper is to provide a framework for investigating the important convective and convectively-coupled processes over the Maritime Continent, and how they are represented in models. It is well-known that important differences arise in the relationship between convection and SST on intraseasonal timescales between coupled and atmosphere-only simulations (see e.g. Figure 7 of Kim et al. (2010)). In atmosphere-only models on intraseasonal timescales convection tends to become in phase with SST as a result of the higher boundary layer moist static energy, whereas in coupled simulations the high SSTs are associated with periods of clear skies and low windstresses which lead to ocean warming. In the MJO this leads to an observed quadrature between convection and the SST in observations and coupled models compared to a more in phase relationship between convection and SST in atmosphere only models. We therefore do not intend to demonstrate that the mean-state and variability produced by the coupled runs are *better* than the same atmospheric model runs with prescribed SSTs when compared to observations; rather, we intend to demonstrate that coupled runs produce realistic mean-state and variability.

   To demonstrate the robust qualitative difference in the phase relationship, we will add a second panel to Figure 13, showing the grid-point lead-lag relationship between precipitation and SST for the 2015-16 season atmosphere-only runs, compared to the same season for the coupled runs (see attached Figure 3). The correlations are weaker at all times in both MC2 and MC12 for the atmosphere-only run. Perhaps more importantly, the shape of the lead-lag relationship is different, with positive correlation at lead/lag 0, compared to approximately zero correlation in the coupled runs. This is in line with other studies in atmosphere-only models (with a fairly symmetric relationship about lag 0 since the SSTs cannot respond to the precipitation) and coupled runs (where higher SSTs promote the development of higher rainfall, but the precipitation and associated increased cloud cover then cool the SSTs). We note that in this configuration with both the boundary conditions and the prescribed SSTs coming from observations the relationship between SST and convection is perhaps more constrained to be close to the observed relationship than in a free running simulation, but even within that constraint the differences between the coupled and atmosphere-only simulations are large.

   We have re-worded and re-structured Section 5 on Air-Sea coupling to clarify these aspects (ll. 414–464):

[revised manuscript text omitted]

2. We agree that quantifying the overall domain-mean magnitude of the biases is useful. Therefore we have added a table collecting the domain-averaged mean values and interannual standard deviations of variables discussed in sections 3 and 4, as well as their biases and RMS errors relative to reference datasets. We have made a minor addition to the text at the start of Section 3 to introduce the table (ll. 237–239):

   This section considers the mean state of the atmosphere and mixed-layer ocean across the 10 simulation years; Table 3 summarises the domain-mean values, as well as biases and RMS errors relative to reference datasets, of variables considered in this section.

3. We agree that it is important to investigate the representation of clouds when considering convective processes, although cloud fraction itself is not necessarily the most relevant variable for understanding how the variability is represented in models. For example clouds may brighten leading to increases in reflected shortwave radiation or deepen leading to reductions in outgoing longwave radiation without any change in cloud fraction. Furthermore the definition of cloud fraction in models and observations can vary greatly between individual models (or reanalysis products) and depend strongly on the "observation" used

to define them and so a direct comparison between models and observations is not always helpful. We have however compared both the outgoing longwave and reflected shortwave from both model simulations against observations (see attached Figures 4, 5) and find that in both MC2 and MC12 simulations the biases are broadly consistent with typical GCM biases, and that the biases in MC2 tend to be reduced compared to MC12, suggesting a generally better representation of clouds in MC2 (consistent with a higher-resolution convection-permitting simulation). We have added the following text to Section 3 of the revised manuscript to summarize the above (ll. 279–282):

> As proxies for cloud cover, we also compared the outgoing longwave and reflected shortwave from both model simulations against observations (see Supplementary Figures S3 and S4). We find that in both MC2 and MC12 the biases are broadly consistent with typical GCM biases, and that the biases (and RMS errors) in MC2 tend to be reduced compared to MC12, suggesting a generally better representation of clouds in MC2 (consistent with a higher-resolution convection-permitting simulation).

However, as with all other variables considered, the interannual variability is generally too high for both simulations (see attached Figures 6, 7). We have added the following text to Section 4 of the revised manuscript to reflect this (ll. 369–370):

> In each case, standard deviations are comparable but are generally slightly overestimated in both models compared to the observations (this is also true for the top-of-atmosphere shortwave and longwave radiation fluxes; see Supplementary Figures S5 and S6).

We do not agree with the reviewer that biases in the 2m air temperature are particularly important for the kind of process studies for which we expect these simulations to be used. Moreover, over land, surface air-temperature will be particularly sensitive to orographic height and therefore not easy to compare even between the two models where the orography is different. We have therefore not considered this further.

4. We agree that the style of the paper is largely descriptive. We have added a few more explanatory/exploratory comments (in response to both reviewers), but do not wish to present too much speculation without properly digging in to the causes of differences between the model runs, and between the model runs and observations. The purpose of creating this dataset is to explore precisely these questions of why higher resolution/explicit representation of convection changes e.g. the mean state and variability, including for instance coupling with equatorial waves and the MJO.

   In response to the reviewer's specific queries:

   - On the wet bias being stronger in MC2 than MC12: this was not unexpected, as it is fairly common in convection-permitting simulations (see response to Reviewer 1, Specific Comment 1).
   - On the SST biases being weaker in MC2 than MC12: this was unexpected, as the relaxation runs to determine the KPP flux corrections were only performed for the parametrized convection model at N1280 grid spacing. It would have been entirely reasonable to expect that the SST biases would therefore be worse in MC2.

     A full exploration of the mechanisms behind this difference is beyond the scope of this paper, but a preliminary analysis of the bias in both simulations suggests that the change in SST bias between MC2 and MC12 are broadly consistent with changes in the surface heat flux and these changes are also consistent with changes in the mixed-layer depth. This is especially clear around the islands of the Maritime Continent, with cooler SST and deeper mixed layers associated with lower heat fluxes into the ocean.

     However, the pattern of surface fluxes is complicated. The changes in surface radiation budget are broadly consistent with changes in the corresponding top-of-atmosphere fluxes, with some cancellation between the shortwave and longwave fluxes. The changes in the latent-heat (LH) flux are more complex, noting that the LH flux is defined as positive downwards and as such a positive change indicates decreased evaporation. The sign of the LH flux change is not consistent with it being driven directly by the SST, as the evaporation is generally higher in regions of colder SST. Similarly over large regions away from the islands evaporation is lower, but the surface wind stress is increased. The relative contributions of each component of the surface fluxes varies from region to region.

     To properly understand the differences in the SSTs would require a detailed analysis of the time evolution of the biases and would be regionally sensitive. It would be particularly interesting to look at these processes around the coastlines where it is clear there is a significant change in the representation of the diurnal cycle of convection, which is known to be associated with strong onshore/offshore circulations. This would make a good focus for a future study. We have therefore only added a brief discussion of this in Section 3 of the revised manuscript (ll. 246–252):

That the SST biases are smaller in MC2 was unexpected, as the relaxation runs to determine the KPP flux corrections were only performed for MC12. A full exploration of the mechanisms behind this difference is beyond the scope of this paper, but a preliminary analysis suggests that the changes in SST biases between MC2 and MC12 are broadly consistent with changes in the surface heat flux and these changes are also consistent with changes in the mixed-layer depth (not shown). This is especially clear around the islands of the Maritime Continent, with cooler SST and deeper mixed layers associated with lower heat fluxes into the ocean. To properly understand the differences in the SST biases would however require a detailed analysis of the time evolution of the biases and would be regionally sensitive.

[Figure]

Figure 3: Grid-point lead-lag relationship between precipitation and SST averaged across ocean grid-cells between 15° S and 3 ° N. Each season was linearly detrended before computation to remove seasonal and interannual variability. Coloured lines indicate observations in blue (OSTIA compared to GPM-IMERG), MC12 in orange and MC2 in green. Panel a) shows the lead-lag relationship for both coupled simulations suites and observations for all simulation years; panel b) shows the lead-lag relationship for the coupled (solid lines) versus atmosphere-only (dashed lines) simulations for the 2015-16 season only. This figure appears as Figure 13 in the revised manuscript.

[Figure]

Figure 4: Mean outgoing longwave radiation and biases (W m$^{-2}$). Subplots along the diagonal indicate MC2, MC12, and NOAA daily OLR respectively. Upper off diagonal subplots show difference plots between each of the datasets. This figure appears as Supplementary Figure S3 in the revised manuscript.

[Figure]

Figure 5: Mean outgoing shortwave radiation and biases (W m$^{-2}$). Subplots along the diagonal indicate MC2, MC12, and NCEP-NCAR reanalysis respectively. Upper off diagonal subplots show difference plots between each of the datasets. This figure appears as Supplementary Figure S4 in the revised manuscript.

[Figure]

Figure 6: Interannual standard deviation of outgoing shortwave radiation and biases (W m$^{-2}$). Subplots along the diagonal indicate MC2, MC12, and NOAA daily OLR respectively. Upper off diagonal subplots show difference plots between each of the datasets. This figure appears as Supplementary Figure S5 in the revised manuscript.

[Figure]

[Figure]

Figure 7: Interannual standard deviation of outgoing shortwave radiation and biases (W m$^{-2}$). Subplots along the diagonal indicate MC2, MC12, and NCEP-NCAR reanalysis respectively. Upper off diagonal subplots show difference plots between each of the datasets. This figure appears as Supplementary Figure S6 in the revised manuscript.

| Variable | Reference dataset | Model | Mean | Bias | RMSE | Std. dev. | Bias | RMSE |
|---|---|---|---|---|---|---|---|---|
| Precipitation [mm/day] | GPM-IMERG | MC2 | 8.43 | 1.1 | 3.2 | 3.01 | 0.466 | 1.47 |
| | | MC12 | 8.5 | 1.2 | 2.97 | 2.42 | -0.112 | 1.34 |
| | MSWEP | MC2 | 8.43 | 1.35 | 3.41 | 3.01 | 1.25 | 1.77 |
| | | MC12 | 8.5 | 1.45 | 3.01 | 2.42 | 0.67 | 1.33 |
| SST [°C] | OSTIA | MC2 | 28.8 | -0.0188 | 0.144 | 0.337 | 0.0296 | 0.0867 |
| | | MC12 | 28.8 | 0.0231 | 0.205 | 0.349 | 0.0421 | 0.0973 |
| TOA outgoing longwave [W/m$^2$] | NOAA daily OLR | MC2 | 225 | 0.19 | 8.06 | 11.4 | 1.14 | 2.46 |
| | | MC12 | 225 | -0.2 | 14 | 13.1 | 3.07 | 4.03 |
| TOA outgoing shortwave [W/m$^2$] | NCEP-NCAR reanalysis | MC2 | 121 | -20.8 | 28.1 | 11.5 | 3.43 | 5.16 |
| | | MC12 | 120 | -21.9 | 29.4 | 11.7 | 3.59 | 5.34 |
| U850 [m/s] | ERA5 reanalysis | MC2 | -1.5 | -0.69 | 1.1 | 1.47 | 0.226 | 0.798 |
| | | MC12 | -1.71 | -0.835 | 1.13 | 1.28 | 0.0232 | 0.235 |
| V850 [m/s] | | MC2 | -1.14 | -0.0659 | 0.514 | 0.511 | -0.0383 | 0.328 |
| | | MC12 | -1.14 | -0.0671 | 0.42 | 0.652 | 0.0996 | 0.199 |
| Zonal mean uplift [Pa/s] | | MC2 | -0.0309 | -0.00397 | 0.0192 | 0.00942 | 0.00122 | 0.00452 |
| | | MC12 | -0.0319 | -0.00493 | 0.00841 | 0.00858 | 0.000378 | 0.00183 |
| Zonal mean specific humidity [g/kg] | | MC2 | 5.59 | 0.0571 | 0.188 | 0.184 | -0.00709 | 0.047 |
| | | MC12 | 5.77 | 0.237 | 0.331 | 0.171 | -0.02 | 0.0477 |
| Zonal mean air temperature [K] | | MC2 | 263 | 0.21 | 0.441 | 0.298 | 0.00569 | 0.0329 |
| | | MC12 | 264 | 0.24 | 0.573 | 0.293 | 0.000282 | 0.038 |
| Zonal mean U [m/s] | | MC2 | -3.09 | 0.00449 | 0.923 | 1.12 | 0.00532 | 0.119 |
| | | MC12 | -3.07 | 0.0225 | 1.2 | 1.08 | -0.0398 | 0.188 |
| Zonal mean V [m/s] | | MC2 | 0.124 | 0.0151 | 0.392 | 0.321 | 0.0152 | 0.0725 |
| | | MC12 | 0.138 | 0.029 | 0.576 | 0.315 | 0.00896 | 0.126 |

Table 1: Summary of domain-averaged time-mean values and interannual standard deviations of variables discussed in the text, as well as their respective biases and RMS errors relative to reference datasets. All averages are computed over the MC2 domain only. Averages of zonal-mean fields are calculated for pressures less than 100 hPa. This table appears as Table 3 in the revised manuscript.

[Figure]

Figure 8: Spatial maps of differences between the all-season means of variables linked to the SST. Top row (L–R): sea surface temperature; mixed-layer depth; heat flux into the ocean. Middle row (L–R): downward surface shortwave flux; downward surface longwave flux; downward surface latent heat flux (a positive change corresponds to reduced evaporation). Bottom row (L–R): magnitude of downward surface wind stress; top-of-atmosphere outgoing longwave radiation flux (as an indication of cloud changes); top-of-atmosphere outgoing shortwave radiation flux (as an indication of cloud changes). Differences in downward surface sensible heat fluxes are not shown; their magnitude is less than $\sim 3 \text{ W m}^{-2}$ almost everywhere.

**References**

Beck, H. E., van Dijk, A. I. J. M., Levizzani, V., Schellekens, J., Miralles, D. G., Martens, B., and de Roo, A.: MSWEP: 3-hourly 0.25° global gridded precipitation (1979–2015) by merging gauge, satellite, and reanalysis data, Hydrology and Earth System Sciences, 21, 589–615, https://doi.org/10.5194/hess-21-589-2017, 2017.

[revised manuscript text omitted]